# Identifying and Clustering Counter Relationships of Team Compositions in PvP Games for Efficient Balance Analysis

**Chiu-Chou Lin** *dsobscure@outlook.com*
**Yu-Wei Shih** *yoway0430@gmail.com*
**Kuei-Ting Kuo** *kuo0404@gmail.com*
**Yu-Cheng Chen** *yucheng881111@gmail.com*
**Chien-Hua Chen** *x21530317x@gmail.com*
**Wei-Chen Chiu** *walon@cs.nctu.edu.tw*
*Department of Computer Science*
*National Yang Ming Chiao Tung University, Hsinchu 30010, Taiwan*

**I-Chen Wu** *icwu@cs.nycu.edu.tw*
*Department of Computer Science*
*National Yang Ming Chiao Tung University, Hsinchu 30010, Taiwan*
*Research Center for Information Technology Innovation*
*Academia Sinica, Taipei 11529, Taiwan*

*Reviewed on OpenReview:* *https://openreview.net/forum?id=2D36otXvBE*

## Abstract

**How can balance be quantified in game settings?** This question is crucial for game designers, especially in player-versus-player (PvP) games, where analyzing the strength relations among predefined team compositions—such as hero combinations in multiplayer online battle arena (MOBA) games or decks in card games—is essential for enhancing gameplay and achieving balance. We have developed two advanced measures that extend beyond the simplistic win rate to quantify balance in zero-sum competitive scenarios. These measures are derived from win value estimations, which employ strength rating approximations via the Bradley-Terry model and counter relationship approximations via vector quantization, significantly reducing the computational complexity associated with traditional win value estimations. Throughout the learning process of these models, we identify useful categories of compositions and pinpoint their counter relationships, aligning with the experiences of human players without requiring specific game knowledge. Our methodology hinges on a simple technique to enhance codebook utilization in discrete representation with a deterministic vector quantization process for an extremely small state space. Our framework has been validated in popular online games, including *Age of Empires II*, *Hearthstone*, *Brawl Stars*, and *League of Legends*. The accuracy of the observed strength relations in these games is comparable to traditional pairwise win value predictions, while also offering a more manageable complexity for analysis. Ultimately, our findings contribute to a deeper understanding of PvP game dynamics and present a methodology that significantly improves game balance evaluation and design.

## 1 Introduction

In the dynamic landscape of player-versus-player (PvP) games, team compositions, or "comps," such as hero combinations or decks formed before matches commence, are pivotal (Costa et al., 2019; de Mesentier Silva et al., 2019; Reis et al., 2021). The gaming industry, now approximately a 200 billion US dollar market (Kristianto, 2023), thrives on the diversity and engagement offered by these compositions, reflecting players' individuality and sustaining market competitiveness (Figueira et al., 2018; Fontaine et al., 2019). However,

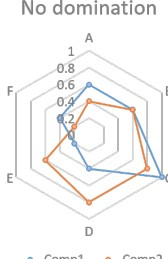 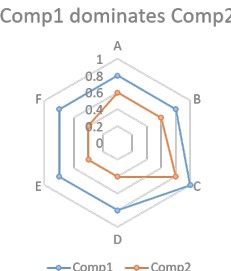

Figure 1: Radar chart comparison of two team compositions across matchups with six different opponents. The left panel illustrates a scenario with no domination, where both compositions exhibit their strengths against specific opponents. The right panel shows a case where Comp1 dominates Comp2, achieving higher win rates against all opponents, illustrating clear dominance in overall performance.

the key to optimizing player engagement and competitive fairness lies in maintaining reasonable strength relations among diverse team compositions—a challenge for both players aiming for victory and designers striving for balance (Levkoff, 2014; Bakkes et al., 2014; Beyer et al., 2016).

A quantitative measure for game balancing is thus essential for addressing this challenge. Currently, win or success rate, use rate, or even the entropy of strategy distributions are available measures across various game genres, targeting optimizations from detailed game parameters to skill-based matchmaking among players (Morosan & Poli, 2017; Hunicke, 2005; Rupp et al., 2023; Nikolakaki et al., 2020; Pendurkar et al., 2023). However, the prevailing reliance on these measures for balance assessment overlooks critical factors such as player skill variability and the counter relationships between compositions, rendering evaluations imprecise. Traditional player skill ratings, including Elo rating, TrueSkill, and Matchmaking Rating, predominantly focus on individual prowess, leaving a gap in the strength assessment of team compositions (Elo, 1966; Herbrich et al., 2006; Pramono et al., 2018).

To better understand strength relations in compositions and analyze game balance, we pose the question: **"How many compositions are not dominated?"** If a composition shows no advantage over others, it could be considered redundant. Hence, our goal in this paper is to define measures that answer this question. We first integrate the Bradley-Terry model with Siamese neural networks (Bromley et al., 1993) to predict the strengths of team compositions from game outcomes under the competitive scenario (Bradley & Terry, 1952; Li et al., 2021). This scalar strength rating helps us identify the strongest or dominating composition more effectively with the numerical max operation compared to the comparison operation over all compositions. However, a single scalar strength often fails to provide precise predictions due to players potentially altering their playstyle under different states (Lin et al., 2021) or the inherent intransitivity present in competitive scenarios (Chen & Joachims, 2016; Balduzzi et al., 2018). Accurate predictions often consider cyclic dominance, such as the Rock-Paper-Scissors dynamic, which the Bradley-Terry model does not capture. By analyzing discrepancies between actual outcomes and Bradley-Terry model predictions, we learn a counter table through neural discrete representation learning (van den Oord et al., 2017), thereby enhancing prediction accuracy and offering insights into counter dynamics without specific game knowledge. During the learning of counter tables, we found that vanilla vector quantization (VQ) training leads to poor codebook utilization (Zhang et al., 2023), especially in small codebook sizes; hence, we proposed a new **VQ Mean Loss** to improve codebook utilization for this new use case. Leveraging these methods, we define new measures of game balance for counting non-dominated compositions that simple win rates face challenges in computation due to high time complexity.

Our contributions are threefold: First, we establish two measures for balance by counting the non-dominated compositions: **Top-D Diversity**, which counts playable compositions given a tolerant win value gap, where the tolerant gap is defined by game designers and can be due to factors like skill or luck that make players willing to play those compositions; and **Top-B Balance**, which considers counter relationships in counting non-dominated compositions, i.e., how many meaningful counter relationships exist in the game. Next, we introduce the learning of composition strength and counter relationships, reducing the space complexity of

analyzing composition strength relations from $\mathcal{O}(N^2)$ to $\mathcal{O}(N+M^2)$, where $N$ is the number of compositions, and $M$ is the category count of the counter table. This reduction in space complexity is crucial not only for storage reasons but also for generating a feasible size of balance report for game designers. Additionally, the time complexity of **Top-D Diversity** shifts from $\mathcal{O}(N^2)$ to $\mathcal{O}(N)$ and **Top-B Balance** from $\mathcal{O}(N^3)$ to $\mathcal{O}(N+M^3)$. To clarify, this time complexity is only for the strength relationship analysis. The time for collecting game records and obtaining the strength prediction models is not included in this complexity as it depends on how many game records the game designers plan to collect within a given time period and does not necessarily increase with the number of compositions. Lastly, the rating and counter relationships derived from learning align with the experiences of human players without requiring specific game knowledge.

We validate our methods across popular online games such as *Age of Empires II*, *Hearthstone*, *Brawl Stars*, and *League of Legends*, demonstrating precision on par with pairwise strength predictions using neural networks and also showcasing better generality compared to tabular statistics. Our methodology not only exhibits broad applicability but also underscores its potential to transform the evaluation and design processes of game balance. Furthermore, we believe these balance measures are not limited to games, as various competitive scenarios—including sports, movie preferences, peer grading, and elections—exhibit intransitivity in comparisons similar to games (Chen & Joachims, 2016). Additionally, the strength measurement of recent large language models (LLMs) also incorporates PvP paradigms (Zheng et al., 2023).

## 2 Game Balance

Game designers are tasked with devising engaging mechanisms and numerical frameworks that enhance player experiences (Schell, 2008). Developing an immersive game loop not only encourages participation but also assists players in forming a mental model of the game's mechanics (Sellers, 2017). Designers often apply the Yerkes-Dodson law to optimize player satisfaction, suggesting an optimal arousal level for peak performance that aligns in-game challenges with player skill progression (Dodson, 1915). This dynamic interaction is crucial for maintaining players in a state of mental flow (Csíkszentmihályi, 1990), where game balance plays a pivotal role in sustaining appropriate levels of difficulty and challenge.

As a critical research field within game design and operations (Schell, 2008; Novak et al., 2012; Sellers, 2017), game balance significantly influences player engagement through diverse strategies and playstyles. It extends beyond mere difficulty adjustments to encompass strategy, matchmaking, and game parameter tuning (Becker & Görlich, 2020). Understanding balance definitions and metrics is vital for effectively addressing these components. Traditional metrics such as win rate, win value difference, and game scores have driven the evolution of game balancing techniques, refining the interplay between game mechanics and player satisfaction (Jaffe et al., 2012; Budijono et al., 2022; Mahlmann et al., 2012).

In PvP scenarios, win value estimation is a common approach, with values often normalized to scales like [0,1] or [-1,1] to simplify payoff calculations between competitors (Budijono et al., 2022). However, calibrating the strength of a composition with win values typically requires comparisons against multiple opponents (Fontaine et al., 2019). While strength rating systems like Elo, TrueSkill, and Matchmaking Rating can identify strength from a single scalar rating and suggest greater strength with higher ratings (Elo, 1966; Herbrich et al., 2006; Pramono et al., 2018), capturing intransitivity or cyclic dominance in scalar ratings is challenging. This necessitates multi-dimensional ratings (Chen & Joachims, 2016; Balduzzi et al., 2018), which reintroduce complexity into balance analysis. Thus, this paper aims to propose a solution that considers intransitivity while maintaining feasible complexity in balance analysis.

Acquiring accurate game data is also crucial for balance analysis, often involving the deployment of rule-based agents during early development phases and integrating human testers later to capture realistic gameplay data. Advances in artificial intelligence, demonstrated by AlphaZero's performance in board games, have enabled learning-based agents to contribute to game balance data collection (Tomašev et al., 2022). Community discussions about strategies also provide valuable insights, often grounded in game theory principles or defining some empirical relationship graphs by humans to explain the game scenarios (Schmitz, 2022; Hernández et al., 2020). Although the entropy of a strategy reaching Nash equilibrium can serve as a measure of strategic balance (Pendurkar et al., 2023), computing Nash equilibrium policies at the game action level in complex games is resource-intensive, posing a challenge for practical application in the game design

loop (Bowling et al., 2015; Perolat et al., 2022). Although a simpler alternative is using the idea of training adversarial agents or exploiters to identify the weaknesses of main playing strategies (Reis et al., 2024; Vinyals et al., 2019), using rule-based agents or human players as the data source for game balancing is usually more affordable and remains the main approach.

With a comprehensive understanding of game balance, this paper focuses on analyzing balance directly through win-lose outcomes from human players and counting the number of meaningful compositions, particularly in two-team zero-sum PvP games. Given the variability of opposing team compositions in matches, our exploration spans multiple game types, from the civilization choices in *Age of Empire II* to hero combinations in *League of Legends*. Our methodology confronts the challenge of cataloging and evaluating an extensive array of possible team compositions, aiming to enhance the understanding and application of game balance. Before introducing our methods, let us formally define the target, "domination", in PvP balance analysis.

**Definition 2.1.** Define Win : $c_1, c_2 \rightarrow w$ as a way to estimate the winner, where $c_1$ and $c_2$ are the compositions of players 1 and 2, respectively, and $w \in \mathbb{R}, w \in [0, 1]$ is the estimated win value.

**Proposition 2.2.** *We say composition $c_1$ dominates $c_2$ over all compositions $c$ if $Win(c_1, c) > Win(c_2, c)$.*

When Proposition 2.2 is true, $c_2$ is considered useless in terms of win values because $c_1$ can perform better than $c_2$ in all cases. If game designers can validate all compositions with Proposition 2.2, they can analyze game balance by identifying overly strong or useless compositions. If some compositions are very weak or meaningless, leading to most compositions being able to defeat them 100% of the time, thus violating the domination relation, designers can either manually eliminate these compositions from the set of all compositions $c$ or iteratively eliminate dominated compositions by running Proposition 2.2 several times.

However, the time complexity of validating Proposition 2.2 is $\mathcal{O}(N^3)$ over $N$ team compositions with a pairwise win value estimation $\text{Win}(c_1, c_2)$. We will try to reduce this complexity with approximations later.

## 3 Learning Rating Table and Counter Table

For understanding the strength relations between compositions (comps) and performing efficient balance analysis, we need to quantify the strength and counter relationships first. Our methodology begins with the application of the Bradley-Terry model to allocate a scalar value representing the strength of each comp based on win estimations. This process is elaborated upon in Section 3.1. To tackle the issue of cyclic dominance or intransitivity of win values efficiently, epitomized by the Rock-Paper-Scissors dynamic, we devise a counter table. This involves examining the variances between actual win outcomes from specific comps and the predictions made by the Bradley-Terry model, a process detailed in Section 3.2. Furthermore, the overarching framework that integrates these components into our learning process is delineated in Section 3.3.

### 3.1 Neural Rating Table

Win rates in PvP games, while useful as a conventional metric, do not fully encapsulate the actual strengths of individual players or team compositions. A player's or comp's true prowess is better reflected in their ability to triumph over comparable opponents, as victories against both weaker and stronger opponents contribute equally to the win rate but signify different levels of strength. The Elo rating system, commonly utilized in chess and similar two-player zero-sum games, offers a scalar strength rating for entities, aligning with the principles of the Bradley-Terry model (Elo, 1966; Bradley & Terry, 1952). This model predicts the probability of player $i$ defeating player $j$, as delineated in Equation 1:

$$P(i > j) = \frac{\gamma_i}{\gamma_i + \gamma_j}, \tag{1}$$

where $\gamma_x$ represents the positive real-valued strength of player $x$. To manage the scale of $\gamma$, it is often reparameterized using a rating value $\lambda$ in an exponential function, as shown in Equation 2:

$$P(i > j) = \frac{e^{\lambda_i}}{e^{\lambda_i} + e^{\lambda_j}}. \tag{2}$$

Adopting this model, we treat comps analogously to individual players, estimating each comp's strength to predict win probabilities. Given the impracticality of analyzing an extensive $N \times N$ win rate table for a large number of comps, we harness the Bradley-Terry model in conjunction with neural networks to overcome this challenge. Our approach employs a Siamese neural network architecture to deduce the ratings $e^\lambda$ for each comp, utilizing mean square error (MSE) as a regression loss function $D$ for model approximations. In our early experiments, we tried using binary cross entropy as the loss function for its probabilistic nature. However, this approach encouraged the rating values to become very large and prevented the model from converging, similar to using hinge loss. Therefore, we focused on using MSE for stable training.

This integration allows our neural network to learn the rating table $R_\theta$ from match outcomes, assigning a rating to each comp $c$ through $R_\theta(c)$. The ratings are computed using an exponential activation function to ensure appropriate scaling. The loss function, focused on match outcome $W_m$, is formalized as follows:

$$L_{R_\theta} = \mathbb{E}[D(W_m, \frac{e^{\lambda_{m_i}}}{e^{\lambda_{m_i}} + e^{\lambda_{m_j}}})] = \mathbb{E}[D(W_m, \frac{R_\theta(c_{m_i})}{R_\theta(c_{m_i}) + R_\theta(c_{m_j})})]. \tag{3}$$

By adopting this methodology, our network efficiently processes diverse comp combinations, offering a robust and scalable solution for predicting team composition strengths. We can efficiently identify the strongest composition by tracing the ratings over all compositions with a time complexity of $\mathcal{O}(N)$.

## 3.2 Neural Counter Table

Within the framework of adapting the Bradley-Terry model through neural networks, we can list the strength of all $N$ team compositions with a space complexity of $\mathcal{O}(N)$. However, the precision of strength relations provided by this method may not match the precision afforded by direct pairwise comparisons for each composition, a process that inherently bears a space complexity of $\mathcal{O}(N^2)$. While theoretically feasible via neural networks, such direct prediction incurs a high space complexity, making it challenging to check these ratings and analyze balance, especially with a large $N$.

The phenomenon of cyclic dominance or intransitivity of win values, a common challenge in analyzing game balance, introduces further complications. An $N \times N$ counter table, which would record adjustments from rating predictions to enhance accuracy, becomes impractical due to its high space complexity and cognitive load. In practice, players intuitively grasp counter relationships without the need for exhaustive memorization of large tables. To navigate this, we propose a more manageable $M \times M$ counter table that serves as an approximation of the full $N \times N$ relationships, where $M$ represents a manageable number of discrete categories. Beginning with a minimum of 3 to capture basic cyclic dominance patterns, $M$ can be adjusted to strike a balance between prediction accuracy and table interpretability.

For the task of learning discrete categories, we employ Vector Quantization (VQ), a technique of neural discrete representation learning celebrated for its effectiveness (van den Oord et al., 2017). It acts as an end-to-end analog to K-means clustering within neural networks, primarily introduced in the context of VQ-VAE (van den Oord et al., 2017; Baevski et al., 2020). Our goal diverges from traditional autoencoder objectives; rather than reconstructing inputs, our focus is on developing a counter table that learns from the residuals between direct win predictions and those derived from the Bradley-Terry model.

### 3.2.1 Neural Discrete Representation Learning

Before introducing our design for learning the counter table, we first discuss a popular discrete representation learning method with neural networks, VQ-VAE (van den Oord et al., 2017). In scenarios that require discrete representation, clustering is a common approach, with k-means clustering being a widely used method for several decades (MacQueen et al., 1967). This clustering idea is based on finding $k$ reference points in the feature space to represent corresponding groups of actual features using the nearest neighbor method. However, obtaining an effective feature space from raw observations for this clustering process is a critical problem. With the growth of deep learning, variational autoencoder (VAE) (Kingma & Welling, 2014) was proposed to learn effective latent feature spaces for several tasks.

Building on the ideas of autoencoders and k-means clustering, VQ-VAE uses the concept of preparing an embedding codebook and employing the nearest neighbor method to convert continuous latent features into the discrete indexes of the codebook, thereby obtaining the discrete representation. Afterward, we can restore the discrete representations back to continuous for decoding tasks. This discretization process can be formulated with the following notations: Given an observation $o$ and its latent features $z_e$ through encoder layers and an embedding space $E = e_1, \cdots, e_K$ with size $K$ (codebook), we can define the following probability function for mapping $o$ to discrete space:

$$q(\overline{s} = k|o) = \begin{cases} 1 & \text{for } k_i = \arg\min_{j} ||z_e^i(o) - e_j||^2, \\ 0 & \text{otherwise.} \end{cases} \tag{4}$$

Through this mapping, we can train a discrete encoder in an end-to-end manner without the need for feature engineering first as in k-means clustering.

For training this neural network, we use the following loss functions:

$$L_{rec} = \mathbb{E}[D(o, o')], \quad L_{vq} = \mathbb{E}[D(sg[z_e], z_q)], \quad L_{commit} = \mathbb{E}[D(z_e, sg[z_q])] \tag{5}$$

Here, $D$ is a distance function, with mean square error (MSE) being a common choice. $z_e$ and $z_q$ are the latent codes before and after nearest neighbor replacement, respectively. The $sg[\cdot]$ denotes the stop-gradient operator. The standard VQ-VAE minimizes the combined loss function $L = L_{rec} + L_{vq} + \beta \times L_{commit}$, where $\beta$ is a weight term that encourages the encoder to produce latent codes closer to discrete representations.

For applying this loss to the encoder, we can use the gradient copy trick with the chain rule, as follows:

$$\nabla\theta_{encoder} = \frac{\partial L_{rec}}{\partial z_q} \times \frac{\partial z_e}{\partial\theta_{encoder}} + \beta \times \frac{\partial L_{commit}}{\partial\theta_{encoder}} \tag{6}$$

For applying this loss to the codebook, it is treated as a simple regression optimization problem.

### 3.2.2 Applying Vector Quantization to the Counter Table

After a brief understanding of vector quantization with neural networks, we extend this idea of discrete representation to our counter table application. Given the symmetrical nature of residual win values and our aim to classify compositions into $M$ discrete categories, we utilize Siamese network architectures for both the learning of discrete representations and the prediction of residual win values, as illustrated in Figure 2. The residual win value, $W_{res}$, is defined as:

$$W_{res}(c_{m_i}, c_{m_j}|R_\theta) = W_m - \frac{R_\theta(c_{m_i})}{R_\theta(c_{m_i}) + R_\theta(c_{m_j})} \tag{7}$$

The counter table, denoted as $C_\theta$, comprises a discrete encoder $Ce_\theta$ and a residual win value decoder $Cd_\theta$, functioning as follows:

$$C_\theta(c_{m_i}, c_{m_j}) = Cd_\theta(Ce_\theta(c_{m_i}), Ce_\theta(c_{m_j})). \tag{8}$$

Here, every output of $Ce_\theta(x)$ belongs to $Ck_\theta$, the embedding space optimized for vector quantization.

The core loss function focuses on the residual win values:

$$L_{res} = \mathbb{E}[D(W_{res}(c_{m_i}, c_{m_j}|R_\theta), C_\theta(c_{m_i}, c_{m_j}))] \tag{9}$$

complemented by a vector quantization loss:

$$L_{vq} = \mathbb{E}\left[\frac{D(z_e(c_{m_i}), z_q(c_{m_i})) + D(z_e(c_{m_j}), z_q(c_{m_j}))}{2}\right] \tag{10}$$

where $z_e$ represents the latent code before vector quantization, and $z_q$ denotes the code post-quantization.

In our application, we require a minimal discrete state space to learn the counter table effectively. We observed low utilization of vectors within the embedding space $Ck_\theta$, leading to unselected vectors during training, which cannot construct a comprehensive $M \times M$ counter table. This low codebook utilization problem is common in VQ, and there are many techniques to improve it (van den Oord et al., 2017; Yu et al., 2022; Shin et al., 2023). Reg-VQ Zhang et al. (2023) specifically discusses this codebook utilization problem and suggests adopting KL divergence in stochastic VQ and leveraging Gumbel sampling over convolutional feature blocks. However, this raises the question of whether there is a simple and easy way to guarantee codebook utilization improvement conceptually and can be easily implemented for the vanilla VQ process for our simple usage. To handle this, we propose a new loss term for embedding vectors, termed VQ Mean Loss, which calculates the distance from the mean vector in the embedding space to the continuous latent code $z_e$. This mechanism can be seen as another K-means clustering, encouraging the vectors in $Ck_\theta$ to gravitate towards $z_e$, thus increasing their likelihood of being selected by the nearest neighbor in subsequent iterations. For a more concrete explanation, we provide an example in Appendix A.1. We define this additional loss as:

$$L_{mean} = \mathbb{E}\left[\frac{D(z_e(c_{m_i}), \overline{e}_k) + D(z_e(c_{m_j}), \overline{e}_k)}{2}\right], \quad \text{where } \overline{e}_k = \frac{1}{M}\sum_{e_k \in Ck_\theta} e_k. \tag{11}$$

The gradients for each component in the counter table learning process are calculated with the hyperparameters $\beta_N$ and $\beta_M$. $\beta_N$ is utilized in VQ-VAE to ensure the continuous latent codes $z_e$ closely align with their quantized versions $z_q$, and we use $\beta_M$ to activate $L_{mean}$, respectively:

$$\nabla Ce_\theta = \frac{\partial L_{res}}{\partial z_q} \times \frac{\partial z_e}{\partial Ce_\theta} + \beta_N \times \frac{\partial L_{vq}}{\partial Ce_\theta}, \ \nabla Ck_\theta = \frac{\partial L_{vq}}{\partial Ck_\theta} + \beta_M \times \frac{\partial L_{mean}}{\partial Ck_\theta}, \ \nabla Cd_\theta = \frac{\partial L_{res}}{\partial Cd_\theta}. \tag{12}$$

By employing this $M \times M$ counter table, win values $W_\theta$ that consider counter relationships can be computed via Equation 13:

$$W_\theta(c_{m_i}, c_{m_j}) = \frac{R_\theta(c_{m_i})}{R_\theta(c_{m_i}) + R_\theta(c_{m_j})} + W_{res}(c_{m_i}, c_{m_j}). \tag{13}$$

This approach reduces the space complexity of analyzing strength relations for $N$ compositions from $\mathcal{O}(N^2)$ to $\mathcal{O}(N + M^2)$. When $M$ is a small, constant value, the complexity can simplify further to $\mathcal{O}(N)$.

### 3.3 Learning Procedure

The methodology underlying the construction of the rating and counter tables is encapsulated in the learning framework depicted in Figure 3. This framework requires a dataset consisting of match results, including the team compositions of the competing sides alongside the ultimate win-lose outcomes. The representation of these compositions is adaptable, ranging from simple binary encodings to more nuanced feature descriptions, according to the preferences and requirements set by game designers.

Crucially, the derivation of the Neural Counter Table $C_\theta$ is predicated on the prior establishment of the Neural Rating Table $R_\theta$. This sequential approach ensures that the foundational ratings of team compositions are accurately determined before their interrelations and counter dynamics are analyzed. The development and refinement of these tables pave the way for the introduction of novel measures aimed at enhancing diversity and balance within the gaming environment. A comprehensive discussion of these newly introduced balance measures is forthcoming in Section 5, while the effectiveness and precision of the rating and counter tables will be evaluated in Section 4.

## 4 Accuracy of Strength Relations

With our rating table and counter table, we can approximate the win value of a match given two compositions and identify the strength relations for balance. In this section, we examine the accuracy of strength relations using these tables across different games and investigate the impact of the hyperparameters $\beta_N$ and $\beta_M$ on counter table training. There are 5 models for each method in our experiments, each trained from a different random seed. The results in the tables are the average values of these models.

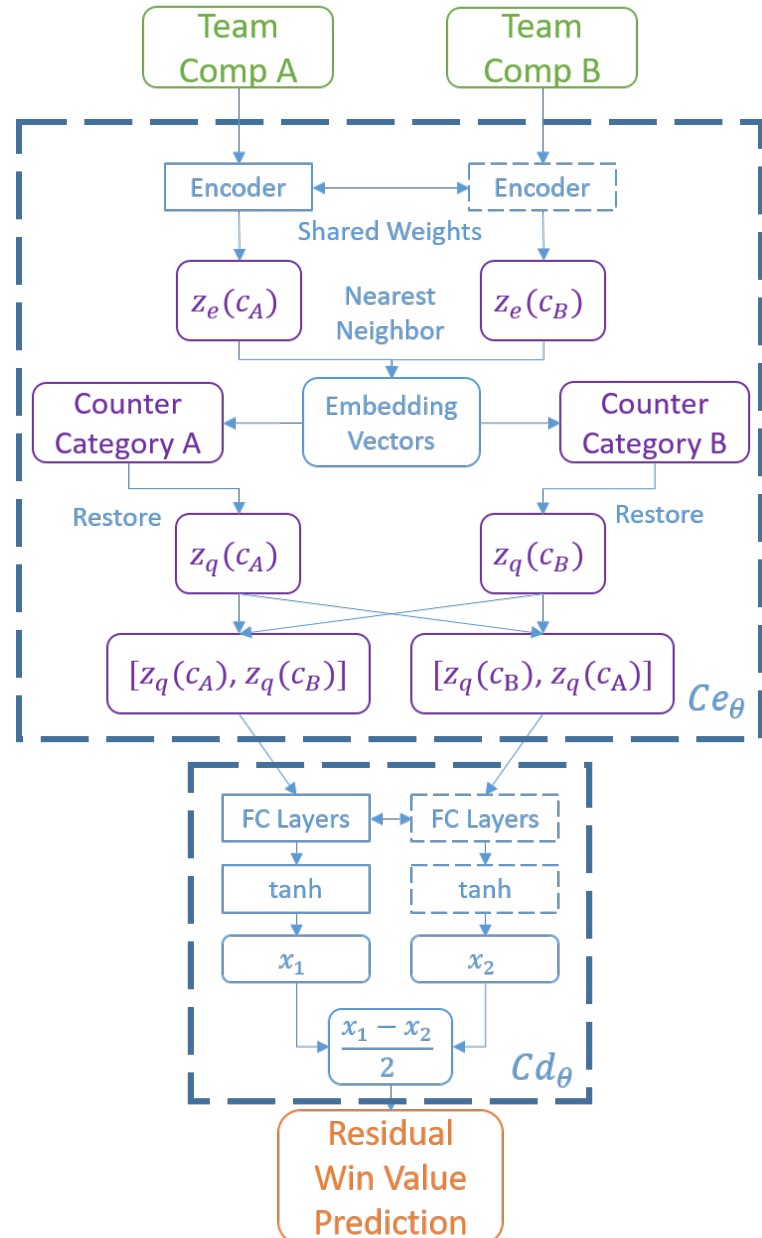

Figure 2: Architecture of the Neural Counter Table $C_\theta$. The diagram illustrates the process of estimating residual win values between team compositions. Team Comp A and Team Comp B are encoded into latent representations $z_e(c_A)$ and $z_e(c_B)$ through shared encoder weights. These latent codes are then quantized into embedding vectors $z_q(c_A)$ and $z_q(c_B)$ using the nearest neighbor search. The embedding vectors are classified into counter categories A and B. The decoded quantized vectors $[z_q(c_A), z_q(c_B)]$ and $[z_q(c_B), z_q(c_A)]$ are fed into fully connected (FC) layers with tanh activation functions to produce intermediate values $x_1$ and $x_2$. The residual win value prediction is calculated as the average of the differences between these intermediate values, providing an estimation of the residual win value $W_{res}$. The dashed layer implies shared weights.

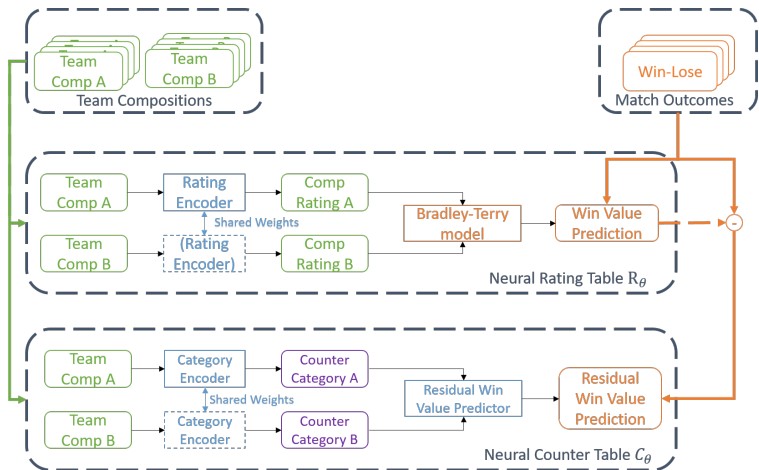

Figure 3: This diagram illustrates the learning procedure for deriving the Neural Rating Table $R_\theta$ and the Neural Counter Table $C_\theta$. The process begins with team compositions (Team Comp A and Team Comp B) and their corresponding match outcomes (win-lose results). In the first stage, team compositions are processed through a shared rating encoder to obtain composition ratings (Comp Rating A and Comp Rating B). These ratings are then utilized in the Bradley-Terry model to predict win values, forming the Neural Rating Table $R_\theta$. In the second stage, these compositions are further processed through a shared category encoder to determine counter categories. The residual win value predictor uses these categories to refine win value predictions, accounting for cyclic dominance, thus forming the Neural Counter Table $C_\theta$.

## 4.1 PvP Games

To assess the accuracy of strength relations with our tables, we constructed simple games that emulate practical game scenarios for experiments. Additionally, we applied our methods to several open-access e-sports game records to confirm their real-world applicability.

### 4.1.1 Simple Combination Game

A simple combination game was designed with 20 elements, each assigned a score equal to its index from 1 to 20. A comp consists of three distinct elements. The score $s_c$ of a comp $c$ is the sum of its elements' scores, and the win-lose outcome is binary, sampled via the probability function $P(c_1 > c_2) = \frac{(s_{c_1})^2}{(s_{c_1})^2 + (s_{c_2})^2}$. **There are $C_3^{20} = 1140$ possible compositions in this game.** The dataset, comprising 100,000 matches, was generated by uniformly sampling two comps.

### 4.1.2 Rock-Paper-Scissors

We adhered to the Rock-Paper-Scissors rules, **only 3 compositions** with win values of 0/0.5/1 for lose/tie/win, respectively. The dataset, consisting of 100,000 matches, generated by uniformly sampling.

### 4.1.3 Advanced Combination Game

This game combines the simple combination game with Rock-Paper-Scissors rules. The primary rule mirrors the simple combination game, with an additional rule: $T = s_c \mod 3$, assigning $T$ as the counter category of the comp, with 0/1/2 corresponding to Rock/Paper/Scissors. A winning Rock-Paper-Scissors comp receives a +60 score bonus during comp score calculation. **There are still $C_3^{20} = 1140$ possible compositions in this setting.** The dataset consists of 100,000 matches generated uniformly.

### 4.1.4  Age of Empires II (AoE2)

Age of Empires II is a popular real-time strategy game. We utilized the statistics (as of January 2024) from aoestats[1], an open-access statistics website. The game features 45 civilizations (comps) in 1v1 random map mode across all Elo ranges. **Thus, there are 45 compositions in this game. Further combinations of civilizations in team mode or specific maps are not discussed in this paper.** The dataset contains 1,261,288 matches.

### 4.1.5  Hearthstone

For Hearthstone, a popular collectible card game, we accessed the statistics (as of January 2024) from HSReplay[2], an open-access statistics website. We considered 91 named decks as compositions for standard ranking at the Gold level. **Therefore, only up to 91 compositions were used, and the detailed hero or card selections within these compositions were not considered for simplicity.** The dataset comprises 10,154,929 matches.

### 4.1.6  Brawl Stars

Brawl Stars is a popular Multiplayer Online Battle Arena (MOBA) game. We focused on the Trio Modes of Brawl Stars, where teams of three compete for victory. Data were sourced from the "Brawl Stars Logs & Metadata 2023" on Kaggle, initially collected via the public API. With 64 characters, 43 maps, and 6 modes, the composition count could reach $C_3^{64} \times 43 \times 6$, **i.e., the maximum number of compositions can reach 10,749,312**. The dataset includes 179,995 matches, with 94,235 unique compositions observed.

### 4.1.7  League of Legends

For League of Legends, a renowned MOBA game with 5-on-5 team competition, we used the "League of Legends Ranked Matches" dataset from Kaggle, which features 136 champions. **Thus, the maximum number of compositions is $C_5^{136} = 359,933,112$.** The dataset covers 182,527 ranked solo games with 348,498 unique compositions observed, which implies that almost all compositions are different.

## 4.2  Comparisons of Strength Relation Prediction

To better understand whether win value predictions can provide accurate strength relations, we examine the accuracy of the strength relation classification task (weaker/same/stronger) rather than the prediction error of win values. For example, if there are two compositions, A and B, and the oracle win value is Win(A,B)=0.55, the actual outcome we care about is whether A is stronger than B. Now, consider two approximations: Win'(A,B)=0.49 and Win"(A,B)=0.62. It is clear that Win' has a smaller absolute error (0.06) compared to Win" (0.07), but Win' suggests the wrong strength relation.

In this strength relation classification task, if the win value falls within the range of [0.499, 0.501], we designate the prediction as the same; a value below 0.499 indicates weaker, and a value above 0.501 indicates stronger. The classification label is calculated based on the average pairwise win value in the dataset. For example, if $C_A$ has a 60% average win value against $C_B$ in the dataset, $C_A$ is deemed stronger when calculating accuracy. All models are approximated with neural networks and trained for 100 epochs on datasets using 5-fold cross-validation. A linear decay learning rate from 0.00025 to 0 over 100 epochs with the Adam optimizer is employed. We then compare the following five methods of win value prediction and provide their definitions with formulations in Section A.4.1:

1. **WinValue**: Predicts the win value for a given composition and compares the win values of the two compositions to determine the winner. If the absolute value of the win value difference is not greater than 0.1%, they are considered to be at the same level. This is a common method in game statistics that does not require maintaining a large table.

---

[1] https://aoestats.io
[2] https://hsreplay.net/

Table 1: Accuracies (%) in training (above) and testing (below) for various games, illustrating the effectiveness of **NCT** with $M = 81$ in achieving high prediction accuracy across different games.

|  | **WinValue** | **PairWin** | **BT** | **NRT** | **NCT** M=81 |
|---|---|---|---|---|---|
| Simple Combination | 64.5 | **71.2** | 63.9 | 64.8 | 66.4 |
| Rock-Paper-Scissors | 51.3 | **100** | 51.3 | 51.3 | **100** |
| Advanced Combination | 57.7 | **83.5** | 56.6 | 57.9 | 79.4 |
| Age of Empires II | 68.7 | **97.3** | 68.7 | 68.7 | **97.7** |
| Hearthstone | 81.1 | **97.8** | 81.4 | 83.4 | **97.4** |
| Brawl Stars | 90.2 | 94.3 | 53.2 | 95.9 | **97.2** |
| League of Legends | 79.6 | 78.9 | 54.0 | 88.2 | **90.9** |
| Simple Combination | 64.7 | 61.8 | **65.5** | 64.9 | 63.9 |
| Rock-Paper-Scissors | 51.1 | **100** | 51.1 | 51.1 | **100** |
| Advanced Combination | 56.5 | 79.1 | 57.5 | 56.5 | **79.7** |
| Age of Empires II | 64.5 | **75.7** | 64.5 | 64.5 | **75.4** |
| Hearthstone | 80.9 | **95.4** | 81.1 | 81.2 | 94.8 |
| Brawl Stars | 79.7 | 82.4 | 53.0 | 82.8 | **83.4** |
| League of Legends | 51.1 | 50.9 | **53.6** | 51.1 | 51.0 |

2. **PairWin**: Directly predicts the pairwise win value. Some game statistics provide this kind of result when the number of compositions is not too large, and it is a straightforward measure to examine counter relationships. If we have sufficient match results, this method represents the upper bound of strength relation accuracy.

3. **BT**: Utilizes linear approximation to perform the Bradley-Terry model. This method assumes the rating of a composition can be derived from the sum of the element ratings within the composition. Common generalized Bradley-Terry models for team setups or Elo ratings in team games use this kind of approach (Coulom, 2007).

4. **NRT**: Employs non-linear approximation to perform the Bradley-Terry model. In many games, combinations of elements in a composition will change the strength of the composition non-linearly.

5. **NCT**: Enhances **NRT** with an additional neural counter table of size $M \times M$.

Table 1 presents the accuracy of the strength comparison task. Notably, **NCT** with $M = 81$ achieves accuracy comparable to **PairWin** across all games. In games with complex compositions, such as Brawl Stars and League of Legends, a non-linear approximation is essential for estimating comp strength. For games with explicit counter relationships—such as Rock-Paper-Scissors, the Advanced Combination Game, and Age of Empires II, which exhibit a significant accuracy discrepancy between **PairWin** and **NRT**—our counter table offers a viable solution. The simplistic win value estimation approach, **WinValue**, usually does not provide the best strength predictions. These results affirm the precision of our rating and counter tables in predicting win values. Notably, in Table 2, as the parameter $M$ increases, so does accuracy, allowing for detailed tracing and analysis of complex counter relationships through the counter table. We provide further discussions regarding the results of counter tables in Appendix A.3.

Additionally, we can observe these accuracies to analyze the properties of the obtained gaming results. If the difference in accuracy between training and testing is small, it implies that there is no significant overfitting and the strength relations can be generalized to matches not observed. However, in cases where there is

Table 2: Accuracies (%) in training (above) and testing (below) for various games with different sizes of counter tables. As the size of the counter table increases, we can obtain better accuracy. Additionally, we can use the difference in accuracy between **PairWin** and **NRT** to identify the magnitude of counter relationships in the game, as these are cases that a single scalar rating system cannot handle.

| | **PairWin** | **NRT** | **NCT** M=3 | **NCT** M=9 | **NCT** M=27 | **NCT** M=81 |
|---|---|---|---|---|---|---|
| Simple Combination | **71.2** | 64.8 | 64.8 | 65.2 | 65.8 | 66.4 |
| Rock-Paper-Scissors | **100** | 51.3 | **100** | **100** | **100** | **100** |
| Advanced Combination | **83.5** | 57.9 | 57.9 | 79.4 | 79.8 | 79.4 |
| Age of Empires II | **97.3** | 68.7 | 68.7 | 73.1 | 83.8 | **97.7** |
| Hearthstone | **97.8** | 83.4 | 81.3 | 85.4 | 91.7 | **97.4** |
| Brawl Stars | 94.3 | 95.9 | 96.3 | **97.3** | **97.2** | **97.2** |
| League of Legends | 78.9 | 88.2 | 89.5 | 91.6 | **92.6** | 90.9 |
| Simple Combination | 61.8 | **64.9** | **64.9** | 64.4 | 64.2 | 63.9 |
| Rock-Paper-Scissors | **100** | 51.1 | **100** | **100** | **100** | **100** |
| Advanced Combination | 79.1 | 56.5 | 56.5 | **79.7** | 80.1 | **79.7** |
| Age of Empires II | **75.7** | 64.5 | 64.5 | 67.7 | 72.5 | **75.4** |
| Hearthstone | **95.4** | 81.2 | 81.3 | 85.2 | 91.3 | 94.8 |
| Brawl Stars | 82.4 | 82.8 | 82.9 | 83.3 | 83.3 | **83.4** |
| League of Legends | 50.9 | 51.1 | 51.1 | 50.9 | 51.0 | 51.0 |

clear overfitting, such as in League of Legends, it suggests the need for more match results to generalize the known strength relations to unknown scenarios. Since League of Legends includes a hero ban and pick phase before a match starts, players tend to prefer compositions that can counter their opponents and aim to improve their win rate. This increases the difficulty of win rate prediction for unobserved cases. The large possible composition space also increases the complexity of accurate model training. For such game cases, we recommend collecting more game results for training. Nonetheless, we can still analyze the strength relations in the training datasets, as our goal is to understand these relations. Initial insights from observed cases can be valuable for early game balancing, although conclusions drawn from these datasets may not necessarily apply to unobserved scenarios.

## 4.3 Counter Table Utilization

Given the need for a counter table for strength relation analysis, we adopt a vector quantization process in our **NCT** training. There is an issue with low codebook utilization when the state space is extremely small. We introduced a VQ Mean Loss to maximize the utilized $M$. For vector quantization, as described in Section 3.2, the standard hyperparameters are set to $\beta_N = 0.01$ and $\beta_M = 0.25$. We explore different configurations of these hyperparameters in Age of Empires II using **NCT** M=27 since it is a game requiring a large counter table for better strength relation accuracy, as shown in Tables 3 and 4. We found that the commonly suggested $\beta_N = 0.25$ (van den Oord et al., 2017) leads to low utilization. Thus, we selected a nearly zero coefficient $\beta_N = 0.01$ to perform regular VQ training. However, even with $\beta_N = 0.01$, if we do not introduce the VQ Mean Loss (set $\beta_M = 0$), the utilized $M$ is still far from the upper bound of 27. We suggest using $\beta_M = 0.25$ for better accuracy and codebook utilization. Also, a coefficient greater than 1 for VQ Mean Loss is not reasonable since it suggests gravitating the mean embedding more than the nearest embedding, which breaks the original idea of VQ and results in worse performance.

Table 3: Training accuracy and the number of utilized categories ($M$) in AoE2 M=27 under different $\beta_N$ with a fixed $\beta_M = 0.25$. The common $\beta_N$ in VQ-VAE is 0.25.

| Setting | Accuracy (%) | Utilized M |
|---|---|---|
| **$\beta_N = 0.01$** | **83.8** | **26.0** |
| $\beta_N = 0.125$ | 71.4 | 5.4 |
| $\beta_N = 0.25$ | 68.7 | 1.0 |

Table 4: Training accuracy and the number of utilized categories ($M$) in AoE2 M=27 under different $\beta_M$ with a fixed $\beta_N = 0.01$. Too small or too large $\beta_M$ cannot provide good codebook utilization.

| Setting | Accuracy (%) | Utilized M |
|---|---|---|
| $\beta_M = 0$ | 76.6 | 13.8 |
| $\beta_M = 0.125$ | 83.6 | 26.0 |
| **$\beta_M = 0.25$** | **83.8** | **26.0** |
| $\beta_M = 0.5$ | 83.4 | 25.6 |
| $\beta_M = 1.0$ | 81.0 | 21.6 |

## 4.4 Tabular Version of Baselines

In Section 4.2, all baselines we used are trained from neural networks for better generalizing unseen compositions. One may ask why not conduct a simple tabular approach like common rating or statistical analysis; thus, we also report the tabular version of WinValue, PairWin, and also Elo rating and a multidimensional variant, mElo2 (Balduzzi et al., 2018).

We test the following five types of methods:

1. **WinValue N→T**: It is the same as **WinValue**, but using a tabular method to get predictions. In other words, this method averages the game results to directly report the average win values instead of using an approximation from a neural network. For those unseen compositions on any side of players in the test dataset, we set the prediction to undefined and always get a wrong prediction since there is no default strength value in the method.

2. **PairWin N→T**: It is the same as **PairWin**, but using a tabular method to get predictions. In other words, this method averages the composition-by-composition game results to give the prediction. For those unseen matches, we also set the prediction to undefined and always get a wrong prediction. We can imagine that this method would have the best training accuracy but very bad generalizability since training with a tabular approximator can have minimal approximation error but no generalization.

3. **Elo N→T**: We apply the standard Elo rating method on compositions. Each composition is a player, and the initial rating is 1000. The constant $K$ for updating the Elo rating is 16. We use NRT as the baseline of Elo since they are all derived from the Bradley-Terry model but use different implementations.

4. **mElo2**: We implement the mElo2 proposed by Balduzzi et al. (2018), which assigns each composition a scalar rating $r$ and a two-dimensional vector $c$. The initial rating is 1000, and the update step $K$ is 16. For the initial vectors of $c$, we follow a public implementation provided by Lazewatsky (2024), uniformly sampled from a real value range [-10, 10].

5. **NCT**: Enhances **NRT** with an additional neural counter table of size $M \times M$.

These tabular methods share the same training process as neural network approaches, including accessing 100 epochs of training data and the random swapping of compositions in a match and corresponding win outcome

Table 5: Accuracies (%) in training (above) and testing (below) with different baselines.

| | WinValue N→T | PairWin N→T | Elo N→T | mElo2 | NCT M=81 |
|---|---|---|---|---|---|
| Simple Combination | 64.5 → 64.5 | 71.2 → **99.9** | 64.8 → 64.3 | 56.6 | 66.4 |
| Rock-Paper-Scissors | 51.3 → 51.3 | **100 → 100** | 51.3 → 73.3 | **100** | **100** |
| Advanced Combination | 57.7 → 57.7 | 83.5 → **99.9** | 57.9 → 57.1 | 52.7 | 79.4 |
| Age of Empires II | 68.7 → 68.7 | 97.3 → **100** | 68.7 → 53.1 | 51.2 | 97.7 |
| Hearthstone | 81.1 → 81.1 | **97.8 → 98.0** | 83.4 → 74.8 | 61.4 | 97.4 |
| Brawl Stars | 90.2 → 95.8 | 94.3 → **99.7** | 95.9 → 98.0 | 97.5 | 97.2 |
| League of Legends | 79.6 → **99.9** | 78.9 → **100** | 88.2 → **100** | 100 | 90.9 |
| Simple Combination | **64.7 → 64.7** | 61.8 → 6.5 | **64.9 → 64.4** | 57.2 | 63.9 |
| Rock-Paper-Scissors | 51.1 → 55.8 | **100 → 100** | 51.1 → 73.6 | **100** | **100** |
| Advanced Combination | 56.5 → 56.7 | 79.1 → 8.2 | 56.5 → 56.1 | 51.3 | **79.7** |
| Age of Empires II | 64.5 → 64.0 | **75.7 → 75.4** | 64.5 → 52.4 | 51.0 | **75.4** |
| Hearthstone | 80.9 → 81.5 | **95.4 → 95.0** | 81.2 → 74.9 | 61.1 | 94.8 |
| Brawl Stars | 79.7 → 69.8 | 82.4 → 69.3 | 82.8 → 77.8 | **83.1** | **83.4** |
| League of Legends | 51.1 → 0.1 | 50.9 → 0 | 51.1 → 6.3 | 50.2 | 51.0 |

adjustment. The results in Table 5 show that tabular methods can have better prediction on training data since they have less approximation error. However, they do not have generalizability and may fall into severe overfitting. If we focus on popular online games, using the neural network version of **PairWin** or **NCT=81** as the win value predictor is still a better choice.

## 5 New Balance Measures

The creation of rating and counter tables allows us to devise new ways to measure balance in games, going beyond simple win rates to consider domination relations as described in Proposition 2.2. In games where two players compete against each other, a common goal is to equalize win rates, aiming for each player to have a win rate near 50%. This is easier to achieve in real-time games with symmetric settings for each player, but it is harder in turn-based games because the player who goes first often has an advantage (Beau & Bakkes, 2016). The main challenge in balancing is determining which player has the upper hand and the extent of their advantage. We propose two new ways to measure balance based on estimated win values and explain how to calculate these measures using our approximations to reduce computational complexity.

Next, we will examine the diversity of comps players might choose in Section 5.1, and identify how many comps might give players an advantage in Section 5.2.

First, let us define some important concepts:

**Assumption 5.1.** The Bradley-Terry rating function $R_\theta(c)$ provides an estimate of Win $= \frac{R_\theta(c_1)}{R_\theta(c_1)+R_\theta(c_2)}$.

**Proposition 5.2.** *The composition $c_{top}$ with the highest rating $R_\theta(c_{top})$ over a rating function $R_\theta$ is considered to dominate all others with lower ratings.*

Considering Definition 2.1, Proposition 2.2, and Assumption 5.1, we can conclude that Proposition 5.2 is true because $\frac{x_1}{x_1+y} > \frac{x_2}{x_2+y}$ when $x_1 > x_2 > 0$. According to Proposition 5.2, if there is only one composition $c_{top}$ with the highest rating, it is considered the best choice before considering counter strategies. This information is often sufficient for some balancing methods, such as identifying and adjusting the strongest comp (Fontaine et al., 2019). The time complexity of identifying $c_{top}$ can be done in $\mathcal{O}(N)$ over $N$ comps,

---

**Algorithm 1** Compute Top-D Diversity Measure

---

**Input:** neural rating table $R_\theta$, top-rated comp $c_{top}$, acceptable win value gap $G$
**Output:** Top-D Diversity Measure $D$
Initialize count $D \leftarrow 0$
**for** each comp $c$ in the set of all comps **do**
    **if** $\frac{R_\theta(c)}{R_\theta(c)+R_\theta(c_{top})} + G \geq 0.5$ **then**
        $D \leftarrow D + 1$
    **end if**
**end for**

---

whereas traditional win rates cannot directly provide a $c_{top}$. If we average the win rates over $N$ comps, the time complexity is $\mathcal{O}(N^2)$. In the following sections, we aim to determine how many compositions might be acceptable to players compared to this top composition and use the new counter table to gain a better understanding of balance through domination with counter relationships.

### 5.1 Top-D Diversity Measure

We are examining how many different game compositions (comps) players might prefer to play. More choices can enrich the game content for fun and also help designers generate revenue by selling these comps. We want to know which comps players will pick based on their chances of winning. Here are some definitions and assumptions for this measure.

**Definition 5.3.** Let us define an acceptable win value gap $G$, where $G \in \mathbb{R}, G \in [0, 1]$.

**Assumption 5.4.** Players think that a small difference in win value, up to $G$, can be attributed to factors like skill or luck, and they are willing to play again under this belief.

**Assumption 5.5.** If a comp $c$ is considered not dominated by $c_{top}$, it is considered not dominated by any comps.

**Lemma 5.6.** *Players are likely to choose comps $c$ where $\frac{R(c)}{R(c)+R(c_{top})} + G \geq 0.5$.*

By accepting Definition 5.3, Assumption 5.1, Assumption 5.4, Assumption 5.5, and Proposition 2.2, Lemma 5.6 is true, since comps that meet this condition are considered not dominated by any other comps. We use Algorithm 1 to count how many comps meet this condition, and this number represents the game's **Top-D Diversity** measure, where a larger $D$ implies more diverse game content. The time complexity of this algorithm is $\mathcal{O}(N)$ over $N$ comps. Without the property of a single scalar rating, the time complexity to check and define win value gaps on pairwise compositions is $\mathcal{O}(N^2)$.

### 5.2 Top-B Balance Measure

To further explore game balance, we recognize that the dynamics of counterplay are vital, and it is rare to have a single dominating comp. Our goal is to identify the number of comps that are not dominated by any other comps, considering their counter relationships.

**Assumption 5.7.** The Bradley-Terry model with the rating function $R_\theta$, enhanced with a counter table $C_\theta$, provides more reliable predictions than using the Bradley-Terry model alone.

Based on Assumption 5.7, we derive the following:

**Proposition 5.8.** *A comp $c_1$ dominates $c_2$ if, for every comp $c$, $Win(c_1, c) + C_\theta(c_1, c) > Win(c_2, c) + C_\theta(c_2, c)$.*

However, verifying this for all comps is still computationally challenging ($\mathcal{O}(N^3)$). To mitigate this, we can categorize comps and record the top comp of each category, leading to the following propositions:

**Proposition 5.9.** *If comps $c_1$ and $c_2$ fall into the same category in $C_\theta$, then $c_1$ dominates $c_2$ when $R_\theta(c_1) > R_\theta(c_2)$.*

**Proposition 5.10.** *If a comp $c_1$ dominates $c_2$ and $c_2$ dominates $c_3$, then $c_1$ also dominates $c_3$.*

---

**Algorithm 2** Compute Top-B Balance Measure

---

**Input:** rating table $R$, counter table $C_\theta$, set of all comps
**Output:** Number of non-dominated comps $B$
List all comps based on $R$ and categorize them using $C_\theta$
Keep only the highest-rated comp in each category
Initialize count $B \leftarrow 0$
**for** each top comp $c$ in each category **do**
    Assume $c$ is not dominated
    **for** each other top comp $c'$ in other categories **do**
        $domi \leftarrow$ true
        **for** each top comp $c''$ in all categories **do**
            **if** $\frac{R(c')}{R(c')+R(c'')} + C_\theta(c',c'') \leq \frac{R(c)}{R(c)+R(c'')} + C_\theta(c,c'')$ **then**
                $domi \leftarrow$ false and break
            **end if**
        **end for**
        **if** $domi$ **then**
            Mark $c$ as dominated and break
        **end if**
    **end for**
    **if** $c$ is not dominated **then**
        $B \leftarrow B + 1$
    **end if**
**end for**

---

Deriving from Proposition 5.8 and Assumption 5.7, Proposition 5.9 simplifies the determination of domination within the same category, and Proposition 5.10 establishes a transitive relationship in domination. These propositions collectively support Lemma 5.11:

**Lemma 5.11.** *Given a rating table $R_\theta$ following the Bradley-Terry model and a counter table $C_\theta$ covering c comps across $M$ categories, all non-dominated comps can be identified among the highest-rated ones in each of the $M$ categories.*

After finding the top comp in each category $(\mathcal{O}(N + M))$, we use Lemma 5.11 to identify non-dominated comps. This methodology is detailed in Algorithm 2, calculating the **Top-B Balance** measure in $\mathcal{O}(N+M^3)$, where a larger $B$ implies more balanced game content.

## 6 Case Study of Top-D Diversity and Top-B Balance

With our new balance measures, previous works on game balancing, as mentioned in Sections 1 and 2, can now incorporate these measures to adjust game mechanisms beyond merely achieving a 50% win rate in PvP scenarios. In this section, we conduct two case studies using our measures for direct balance change suggestions in Age of Empires II and Hearthstone, employing our first model of rating and counter tables in the experiments to demonstrate an application. The actual ratings and counter categories of these tables can be found in Appendix A.3. We also discuss the use case and information for suggesting balance updates with our methods and existing approaches, including win rate observations and entropy-based methods.

### 6.1 Case Study on Age of Empires II

In Age of Empires II, there are 45 civilizations as compositions in 1v1 mode. We first examine the Top-D Diversity measure in Table 6(a). The top composition identified was the **Romans** with a strength of 1.08145. The result on Top-D Diversity suggests that to enhance general balance, setting the win value's standard deviation larger than 4% is reasonable. Such adjustments could be implemented through matchmaking mechanisms, map randomness, game rule variations, etc. A lower randomness level implies imbalance,

Table 6: Top-D Diversity and Top-B Balance in Age of Empires II with different settings.

| (a) | | | (b) | | | |
|---|---|---|---|---|---|---|
| **Setting** | **Top-D Diversity** | | **Setting** | **Used** $M$ | **Top-B Balance** | **Training Accuracy** |
| $G = 0.01$ | 2 | | $M = 3$ | 1 | 1 | 69.6 |
| $G = 0.02$ | 9 | | $M = 9$ | 9 | 8 | 77.0 |
| $G = 0.04$ | 25 | | $M = 27$ | 26 | 24 | 86.0 |
| $G = 0.08$ | 44 | | $M = 81$ | 45 | 45 | 98.9 |

forcing almost every player to choose **Romans**, especially since it is part of the DLC (requiring purchase for competitive advantage).

Regarding Top-B Balance in Table 6(b), when we assume there are 9 categories considering counter relationships, it showed that one category is dominated. According to our ad-hoc analysis in Appendix A.3.1, it is an economic powerhouse category, and the top civilization in this category is **Poles**. This indicates the potential necessity to enhance civilizations within this category on their economic bonuses to improve balance, as even the best among them, **Poles**, is dominated by other top civilizations.

When we extend the size of the counter table to $M = 27$, **Aztecs** and **Chinese** were identified as dominated. This suggests a review of the counter relationships within their category might be warranted to discuss potential improvements. Generally, the balance is commendable, with 24 non-dominated civilizations.

When we assume the counter relationships can be very complex and it is worthwhile for expert players to memorize a large counter table, we find that there is no truly dominated civilization in the $M = 81$ setting. All civilizations are assigned to distinct categories and show their advantages in specific matchups. The accuracy of strength relations in $M = 81$ is 98.9%. This evidence shows that Age of Empires II is balanced when counter relationships are meticulously examined. The game features a complex counter loop, allowing for a counter civilization to almost any other. However, there is still room for balance improvement for beginner players if we do not want such a large counter table.

These insights provide game developers with guidance on addressing balance weaknesses in future updates. They also offer players a deeper understanding that even a generally strong civilization, like Romans, has specific counter civilizations. Traditional game balance techniques, which often aim for a fair win rate (e.g., 50% in 2-player zero-sum games), might overlook the intricacies of counter relationships and game theory when merely weakening strong comps or strengthening weak ones.

## 6.2 Case Study on Hearthstone

When we examine another game, Hearthstone, we first discard decks with fewer than 100 game records in our dataset to ensure the analysis is not biased by outliers. There are 58 decks remaining after our filtering.

From the Top-D Diversity in Table 7(a), the balance is not good since even with a 4% tolerant win value gap, there are only 3 decks considered not dominated. The top deck is **Treant Druid** with a strength of 1.48915 (please see Figure 11 for the strengths of other decks), and it is clearly too strong.

When we check Top-B Balance in Table 7(b) under $M = 3$, **Treant Druid** dominates the others. If we extend the counter table to $M = 9$ (we provide an ad-hoc category analysis in Appendix A.3.2), the balance is surprisingly good with 9 Top-B Balance. This is because there is a clear counter deck, **Aggro Paladin (1.17278)**, to **Treant Druid**. Even though **Aggro Paladin** has a general strength of only 1.17278, it has a strong counter ability to **Treant Druid**, and all top decks in $M = 9$ have their advantages.

When we use a larger counter table, $M = 27$ or $M = 81$, the game is balanced enough since the difference between Used M and Top-B Balance is not significant. This analysis suggests that for this version of Hearthstone, **Treant Druid** requires specific adjustments. The counter relationships are balanced.

Table 7: Top-D Diversity and Top-B Balance in Hearthstone with different settings.

(a)

| Setting | Top-D Diversity |
|---------|-----------------|
| $G = 0.01$ | 1 |
| $G = 0.02$ | 1 |
| $G = 0.04$ | 3 |
| $G = 0.08$ | 9 |

(b)

| Setting | Used $M$ | Top-B Balance | Training Accuracy |
|---------|----------|---------------|-------------------|
| $M = 3$ | 3 | 1 | 81.3 |
| $M = 9$ | 9 | 9 | 87.0 |
| $M = 27$ | 25 | 23 | 90.6 |
| $M = 81$ | 54 | 53 | 97.2 |

### 6.3 Discussion on Different Types of Balance Measures

When considering game balance measures, the major concerns are the type of information these measures provide and how this information can help modify the game mechanics. We focus on measures with fewer subjective factors, specifically those related to improving players' chances of winning. Measures dependent on player preferences, such as use rate, learning difficulty, popularity, and other factors, are not included in the main discussion.

Starting with the most common measure, win rates, it is very clear and general in PvP games. Whether using a simple win rate or a more detailed win rate with specific opponent compositions, making each composition have similar strength is a common idea (Becker & Görlich, 2020). Achieving an average 50% win rate for each composition in an average case without considering its opponents is a specific solution. Designers can check the win rate of each composition and increase the strength of those below 50%, commonly referred to as "buffing" in the player community. For compositions with over 50% win rates, especially those with a large advantage, designers may try to weaken their strength, referred to as "nerfing" in the player community or develop new specific compositions to counter them. However, these changes may not always align with players' desires for entertainment or satisfaction with game depth. Players often pursue diversity or want to demonstrate individuality with different strategy settings (Rheinberg, 2020). Keeping every setting or composition at the same strength level can violate this intention. Therefore, studying different distributions of strength over win rates can provide various solutions, not just achieving 50% average case win rates. Our Top-D Diversity and Top-B Balance measures also rely on win rates but focus on different aspects like the tolerant win value gap and the size of meaningful counter relationships.

Using another possible balance measure, the entropy of strategy, especially strategies that reach Nash equilibrium (Pendurkar et al., 2023), can be considered another application of win rates. A policy reaching Nash equilibrium implies its opponent cannot change itself to gain more benefits. When balance is defined by maximizing the entropy of this policy, it suggests increasing the strength of low sampling probability strategies and reducing the advantages of high sampling probability strategies. This idea is similar to the goal of achieving 50% win rates or the same average case strength but defined on a more complex relationship with Nash equilibrium. However, this idea may have the same limitation as achieving 50% win rates since it cannot explicitly differentiate those compositions that share the same strength relations: for example, making all strategies nearly the same. Therefore, Pendurkar et al. (2023) also proposed using a parameter regularization term to trade off the choice of entropy, which may guide parameters to the same strength, and the inherent diversity of game settings.

Previous measures can help with game balance; however, they cannot guarantee that the updates would not tend to reduce the inherent diversity of the game mechanics since there is usually a global optimal solution that sets all parameters for different compositions nearly the same. Although this result is very balanced, it is also boring for players since there is almost one actual composition decorated as several different compositions or at least sharing the same distribution of win values. If we analyze the potential effect of these balance changes from the player's experience angle, it would converge to a case with not improved balance since the actual game content would be very similar to one strong composition dominating the game since we do not have to study which composition is better, all have the same strength, except the mechanism or playstyle of each composition is different. This raises the question: what result are players pursuing for game balance?

We propose a new explanation: the size of counter relationships. Explicitly increasing the complexity of counter relationships can increase the game's depth, requiring players to study and practice more to master the game with different counter categories to dynamically adjust to the strategies of other players. This ensures a diverse game setting, which is what Top-B Balance can help achieve. Also, accepting the idea of uncertainty from player skill or luck can enrich the game content. If there are weaker compositions, we do not necessarily have to change them to improve balance. Keeping them within a tolerant win value gap allows players to try them occasionally based on the belief in tolerance reasons, increasing the diversity of game mechanics. This is what Top-D Diversity can help with.

There is no clear advantage of one balance measure over another since they can be used simultaneously to provide suggestions. The final decision on which suggestions to use is the responsibility of game designers. Different balance measures provide different information, and we believe our new balance measures can enrich the choices for balance suggestions. Additionally, balance is just one of many factors considered in game design (Schell, 2008). Designers sometimes compromise on balance in favor of more important themes or features that enhance the game experience (Schell, 2008), or players may prioritize playstyle diversity to express their individuality (Lin et al., 2024). To address these kinds of requirements, the regularization term in the entropy-based balance measure (Pendurkar et al., 2023) serves as an example of implementing this trade-off, similar to how the tolerance gap functions in our Top-D Diversity measure.

We also give two simple examples to demonstrate the advantages of our new measures that previous measures cannot explicitly report. Case one: a game without counter relationships, as seen in our Simple Combination Game. There is a theoretically optimal composition: (18,19,20). If we consider the strategy reaching Nash equilibrium, there is only one strategy selecting this composition, and the entropy of this strategy is zero, indicating super imbalance. However, if we consider some slightly weaker compositions: (17,19,20) and (13,17,19), their expected win rates against the best composition are 49.115% and 42.496%, with standard deviations ($\sqrt{p(1-p)}$) of 49.992% and 49.434%, respectively. This shows some probability of reporting (17,19,20) or (13,17,19) as stronger than (18,19,20) with small samples, which may make players feel balanced based on small sample sizes. In this case, Top-D Diversity would help quantify this type of scenario and would not strongly require all compositions to have the same strength, which might need exhaustive effort to develop mechanisms that are not the same but have the same strength.

Another example: we can imagine two extended variants of Rock-Paper-Scissors with payoff matrices in Figure 4. In the original game, we already know that the mixed strategy reaching Nash equilibrium is (1/3,1/3,1/3) and is the uniform distribution. For both the upper variant and the lower variant, the uniform distribution is still a Nash equilibrium solution strategy, and using this strategy would result in a 50% average case win rate. Thus, both entropy-based and simple win rate measures would suggest they are both at the same level of balance. However, when applying our Top-B Balance measure, we can use only a $4 \times 4$ counter table to fully describe the counter relationships in the upper variant (Rock2, Paper2, Scissors2 share the same counter category), but in the lower variant, we need a $6 \times 6$ counter table. This result implies that the lower case has a larger size of counter relationship and players need to study and adjust their strategy for every composition, aligning with our new explanation of improving game balance.

## 7 Conclusion and Future Works

The quantification of balance in competitive scenarios is crucial for analyzing how many participants are meaningful. This paper focuses on a special case of two-team zero-sum competitive scenarios, PvP game compositions. With our approximations of rating and counter relationships, domination relationships can now be quantified efficiently. In the past, most balancing techniques have primarily relied on win rate analysis. Our experiments, conducted in popular online games, underscore the efficacy and practicality of our approach. We believe our work enhances the tools available for game balancing, leading to more balanced and engaging player experiences.

There are still many topics to explore further in the realm of PvP game compositions. For example, we have only considered pre-built compositions, but measuring the balance of elements that form a composition is also important for games with a vast number of element combinations, such as the individual cards in

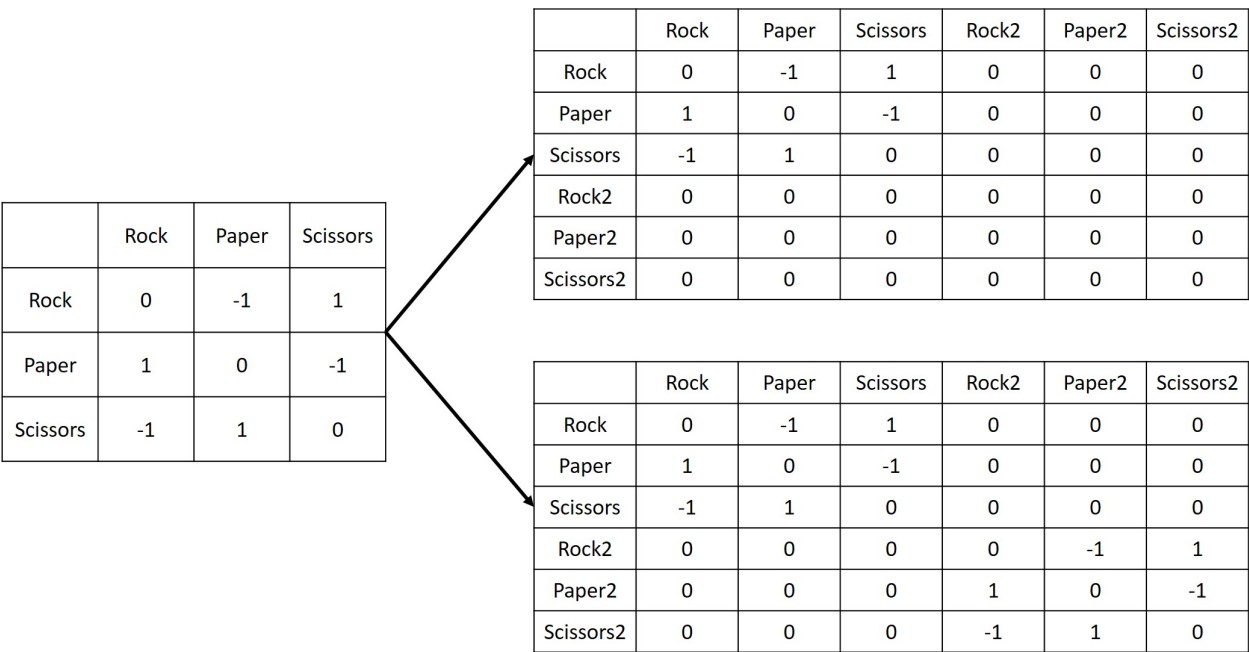

Figure 4: An example of extending the classical Rock-Paper-Scissors to more complex cases.

Hearthstone, the specific tech tree in Age of Empires II, or even the equipment in League of Legends. For games where it is difficult to enumerate all compositions, considering composition building first is essential.

Expanding our focus to broader applications, our approach can be applied to domains where the assessment of competitor strength is crucial in competitive scenarios, such as sports, movie preference, peer grading, elections, and language model agents (Chen & Joachims, 2016; Zheng et al., 2023). In the realm of cutting-edge artificial intelligence research, our approach could offer insights into multi-agent training with counter relationships to exploit weaknesses (Vinyals et al., 2019) and potentially benefit fields like AI safety for attack and defense analysis (Amodei et al., 2016). Thus, our methods hold promise for a wide range of applications, marking a step forward in the quantitative analysis of competitive dynamics.

**Broader Impact Statement**

Our rating and counter tables are learned models and do not guarantee that the results will always be the same. It is necessary to carefully check and train several models for critical applications to ensure the results are not based on random guesses. Our balance measures focus on helping pinpoint the advantages and weaknesses of each composition rather than identifying a single dominating composition. However, if misused, this approach could potentially aid in creating market monopolies rather than improving balance. It is important to use these measures ethically and with the intention of promoting fair competition. Additionally, our methodology is built on a scalar rating system and tested on two-team zero-sum symmetric games, which may limit the applicability of our balance measures to other types of games, especially those without a win rate.

**Acknowledgments**

We appreciate all the valuable feedback and insights from everyone who reviewed this research, which greatly helped in refining and improving this work. Additionally, we would like to thank Gamania Digital Entertainment Co., Ltd. (Gamania) for inspiring the research topics related to game balance in our early studies.

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

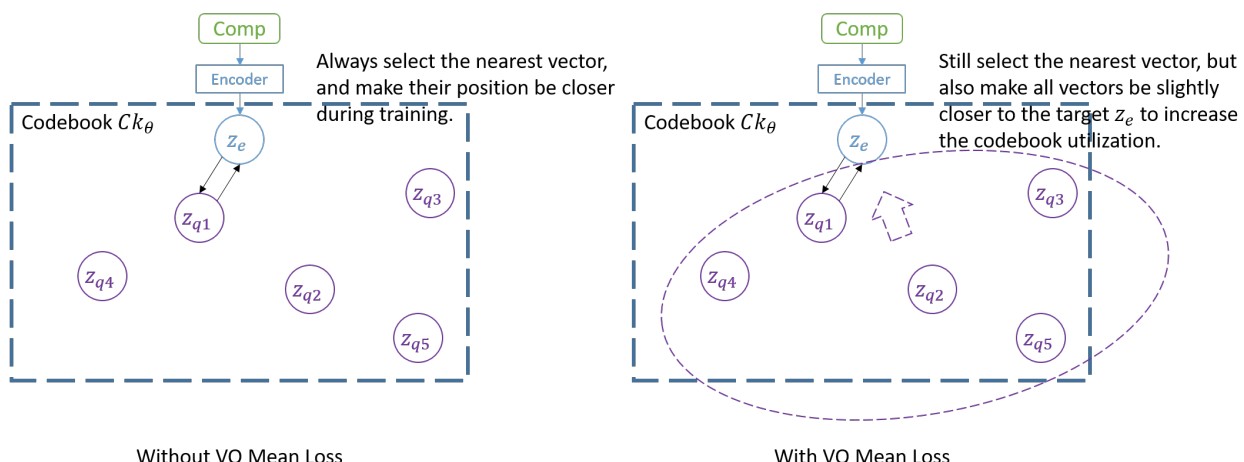

Figure 5: Demonstration of codebook utilization improvement using VQ Mean Loss.

# A    Appendix

## A.1    Enhancing Codebook Utilization Through VQ Mean Loss

We tackle the issue of underutilization in codebooks for Vector Quantization (VQ) by introducing the VQ Mean Loss technique. This method significantly improves the use of the embedding space in VQ models. As depicted in Figure 5, traditional VQ-VAE approaches depend on nearest neighbor selection for determining latent vectors, which can lead to sparse utilization of the codebook. The integration of VQ Mean Loss obviates the need for meticulous codebook initialization or hyperparameter fine-tuning to overcome low utilization problems.

## A.2    Illustrative Examples of Top-D Diversity Measure and Top-B Balance Measure

We elucidate the Top-D Diversity Measure with an example in Figure 6. When utilizing a single scalar for strength measurement of compositions, identifying the composition with maximum strength (denoted as A in the figure) is straightforward. By defining an acceptable win value gap $G$, for instance, $G = 5\%$, which may represent a game's error margin, compositions with a rating equal to or greater than 1.47 are considered playable. Such a demarcation is akin to tier tables commonly created within gaming communities, suggesting that our rating table could facilitate the construction of insightful analyses or tier tables.

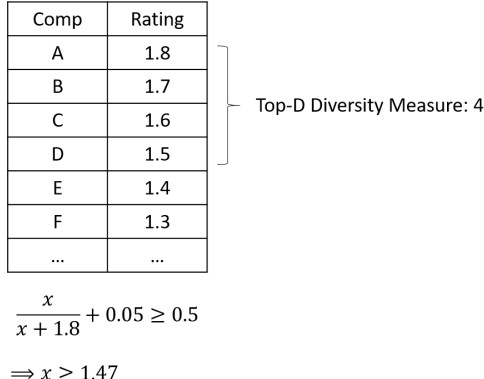

Figure 6: Demonstration of Top-D Diversity Measure, showcasing compositions with ratings above a specified threshold as playable.

For the Top-B Balance Measure, we present a simplified example in Figure 7. Here, compositions are assigned ratings and categorized according to the Rock-Paper-Scissors framework. In this extended gameplay, the highest-rated composition within each category is presumed to be dominant. Therefore, to assess game balance, one only needs to examine the win value relationships between these top compositions. Compositions that are dominated may represent beginner-level or less costly options, requiring players to progress and level up to obtain higher-strength compositions for fair competition ultimately.

| Rock-Comp | Rating | Paper-Comp | Rating | Scissors-Comp | Rating |
|-----------|--------|------------|--------|---------------|--------|
| R1 | 1.7 | P1 | 1.9 | S1 | 1.3 |
| R2 | 1.6 | P2 | 1.4 | S2 | 1.2 |
| R3 | 1.3 | P3 | 1.1 | S3 | 1.1 |
| R4 | 1.2 | P4 | 0.9 | S4 | 1.0 |
| R5 | 1.0 | P5 | 0.7 | S5 | 0.9 |
| R6 | 0.8 | P6 | 0.2 | S6 | 0.8 |
| R.. | .. | P.. | .. | S.. | .. |

Top-B Balance Measure: 3

| Counter | Rock-Comp | Paper-Comp | Scissors-Comp |
|---------|-----------|------------|---------------|
| Rock-Comp | 0 | -50% | +50% |
| Paper-Comp | +50% | 0 | -50% |
| Scissors-Comp | -50% | +50% | 0 |

Figure 7: Explanation of the Top-B Balance Measure, illustrating the assessment of balance by analyzing top compositions within each Rock-Paper-Scissors category.

## A.3 Rating Tables and Counter Tables in Online Games

This appendix provides a discussion of the rating tables and counter tables derived from our first model in Age of Empires II, Hearthstone, and Brawl Stars. Each game's tables reveal interesting insights into the dynamics of team compositions and their interactions. We also explore the rationality and categorization inferred from these tables.

### A.3.1 Age of Empires II

The $9 \times 9$ counter table for Age of Empires II (Figure 8) elucidates the complex interplay of civilization strengths, weaknesses, and counter strategies in the game. With each civilization boasting unique attributes that cater to different playstyles, the table categorizes them into nine distinct groups, reflecting their strategic affinities and shared advantages.

The groups are delineated as follows:

1. **Technological and Age Advancement Group**: Civilizations such as the Malay with accelerated age advancement, the Malians with their fast university research, and the Bulgarians with free militia-line upgrades and significant cost reductions for Blacksmith and Siege Workshop technologies, offering them a strategic edge in pacing the game.

2. **Heavy Cavalry Group**: Notable for their robust cavalry units, civilizations like the Franks and Lithuanians dominate open-field engagements with their superior mobility and combat prowess.

3. **Anti-Cavalry and Anti-Archer Group**: This cohort, including the Goths with their economical infantry and the Indians with their camel units, specializes in countering cavalry and archers, altering the flow of battle with their unique troop compositions.

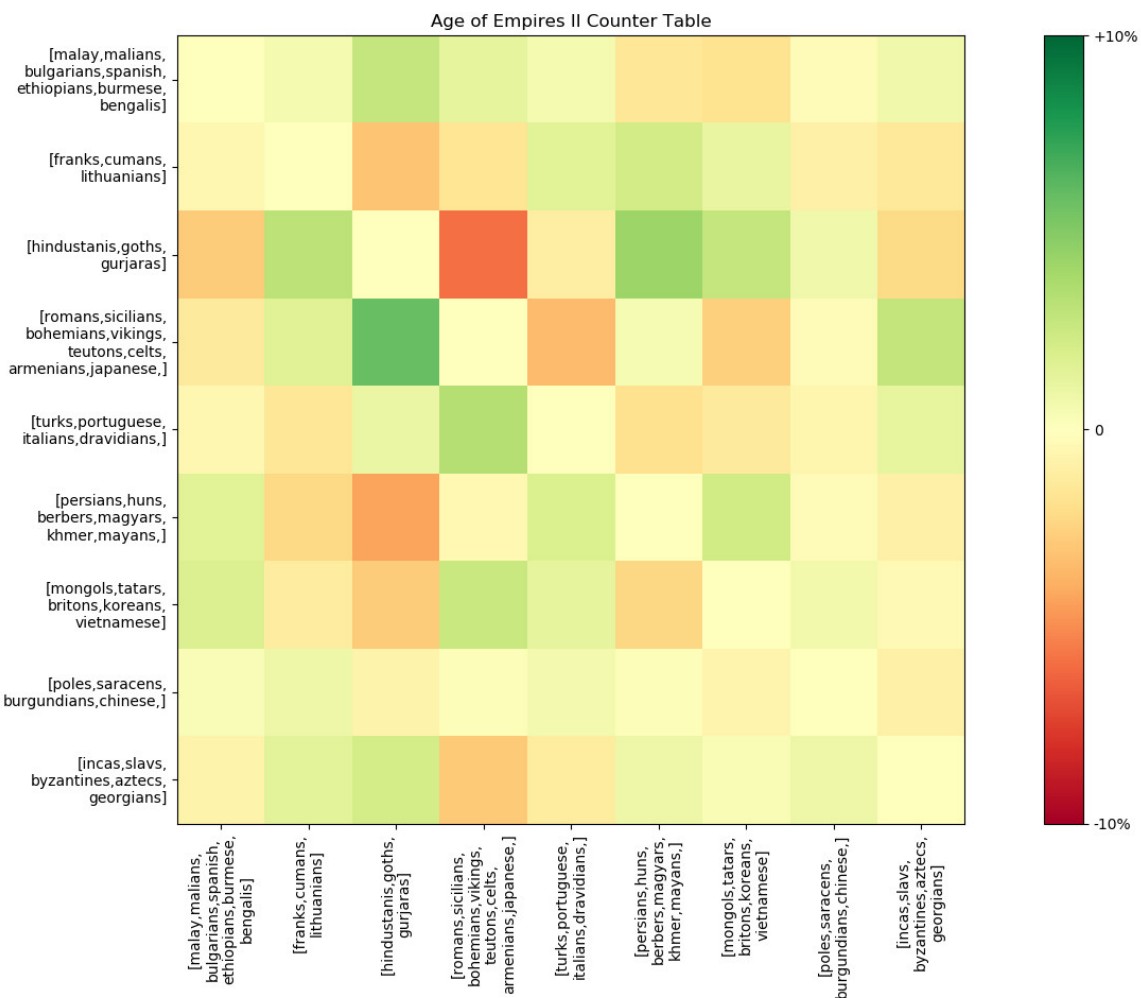

Figure 8: The $9 \times 9$ counter table for Age of Empires II, illustrating the intricate balance of civilization match-ups.

4. **Infantry Dominance Group**: Comprising civilizations with formidable infantry units, this group excels in foot-soldier combat, sustaining front-line engagements and applying pressure through sheer force.

5. **Gunpowder-Intensive Group**: Civilizations in this category leverage the destructive capacity of gunpowder units to gain a decisive advantage in warfare.

6. **Cavalry and Horse Archer Factions**: Balancing strengths between cavalry and horse archers, these civilizations maintain flexible and dynamic combat tactics, adept at swift raids and retreats.

7. **Archery Group**: Dominated by civilizations with powerful ranged units, this group wields archers as the cornerstone of their military strategy, excelling in long-range engagements.

8. **Economic Powerhouses**: These civilizations thrive on economic prowess, underpinning their military campaigns with robust economies and resource accumulation.

9. **Trash and Counter Unit Group**: With a focus on cost-effective 'trash units' and specialized counter units, this group adeptly negates enemy strategies, maximizing efficiency in resource management.

| Category1 | Category2 | Category3 | Category4 | Category5 | Category6 | Category7 | Category8 | Category9 |
|---|---|---|---|---|---|---|---|---|
| Malay 1.00146 | Franks 0.99444 | Hindustanis 1.06402 | Romans 1.08145 | Turks 1.03038 | Persians 0.99886 | Mongols 0.99319 | Poles 0.91685 | Incas 1.02497 |
| Malians 0.97565 | Cumans 0.97313 | Goths 0.99500 | Sicilians 1.02928 | Portuguese 0.91956 | Huns 0.98785 | Tatars 0.83885 | Saracens 0.88485 | Slavs 1.00203 |
| Bulgarians 0.96124 | Lithuanians 0.96764 | Gurjaras 0.93271 | Bohemians 1.02476 | Italians 0.88166 | Berbers 0.97427 | Britons 0.83176 | Burgundians 0.88276 | Byzantines 0.89442 |
| Spanish 0.94718 | | | Vikings 0.99278 | Dravidians 0.87841 | Magyars 0.95044 | Koreans 0.79998 | Chinese 0.85114 | Aztecs 0.87008 |
| Ethiopians 0.94380 | | | Teutons 0.97401 | | Khmer 0.90366 | Vietnamese 0.79796 | | Georgians 0.76448 |
| Burmese 0.87361 | | | Celts 0.94887 | | Mayans 0.88710 | | | |
| Bengalis 0.87242 | | | Armenians 0.91716 | | | | | |
| | | | Japanese 0.89536 | | | | | |

Figure 9: The rating table for Age of Empires II, depicting the overall strength and viability of each civilization.

The rating table (Figure 9) complements the counter table by offering a quantitative assessment of each civilization's overall strength. This allows for a broader perspective beyond direct match-ups, giving insight into how each civilization fares in the general meta.

The counter table not only outlines how civilizations within the same group perform against each other but also illustrates the inherent strengths and weaknesses they possess against other groups, shaped by their distinct technologies, units, and economic bonuses.

For instance, the Heavy Cavalry Group is susceptible to the powerful Anti-Cavalry capabilities of civilizations like the Hindustanis or Goths. They may also find themselves at a disadvantage against Infantry-dominated factions but can leverage their cavalry's mobility to gain an upper hand against civilizations that rely heavily on gunpowder or archery for ranged attacks. Goths, while typically facing a disadvantage in matchups against

other Infantry-focused civilizations, prove to be highly effective against Archery-based groups. It is commonly believed that Infantry-centric civilizations can be countered by those that specialize in gunpowder units and archers; however, they are usually quite efficient against civilizations that rely on 'trash units.' Economically focused civilizations display more flexibility and tend to have less pronounced counter relationships.

Recognizing and understanding these counter dynamics is essential for competitive play. It enables players to predict and neutralize the strategies of their opponents, resulting in more sophisticated and informed decision-making both prior to and during matches. This strategic depth accentuates the long-standing charm of "Age of Empires II" as a competitive real-time strategy game, where tactical knowledge and strategic planning are as crucial as agility and execution.

### A.3.2 Hearthstone

The $9 \times 9$ Hearthstone counter table (Figure 10) represents a strategic breakdown of deck matchups, reflecting how various playstyles interact within the game. Each category is defined by distinct tactical approaches, and the table illustrates the expected performance of these categories against one another, with green indicating a favorable matchup and red indicating a disadvantage.

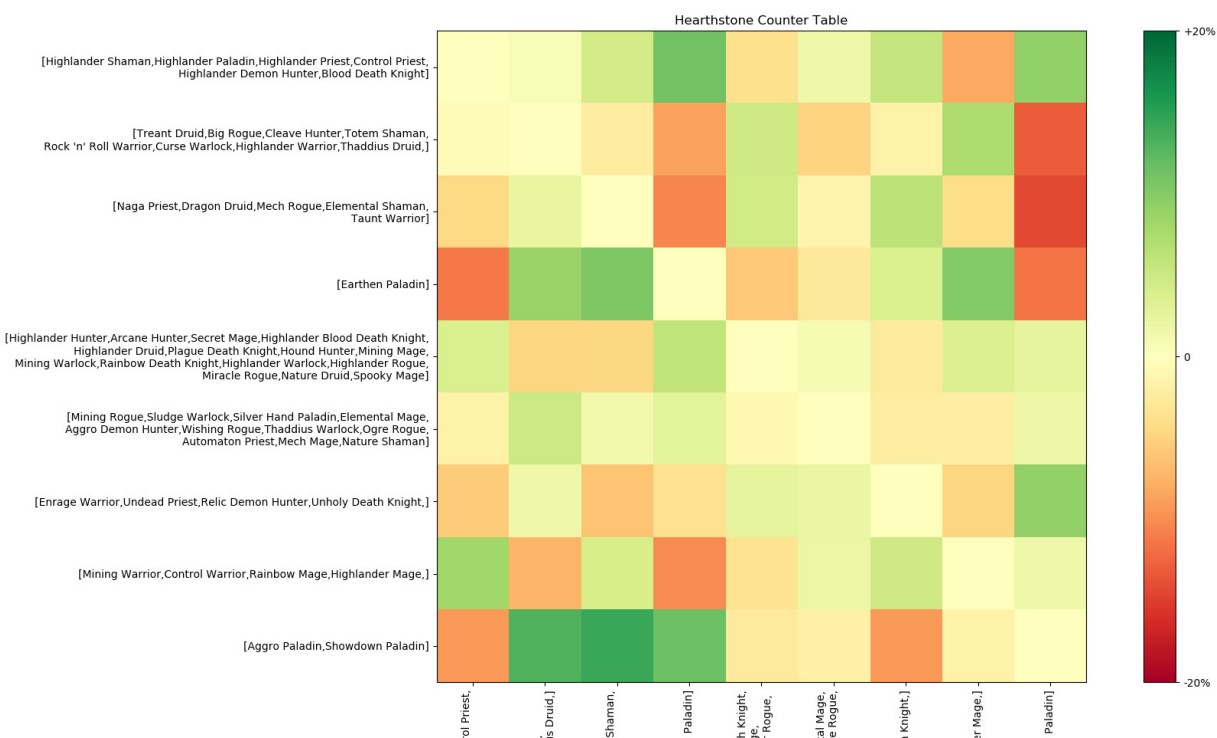

Figure 10: The $9 \times 9$ counter table for Hearthstone, detailing the strengths and weaknesses of various deck archetypes.

1. **High-Value Control Archetypes (Category 1)**: These decks, often Singleton with no duplicate cards, rely on high-value individual cards and powerful board clears, such as the notorious Reno Jackson hero card, to win by value over time. Examples include Highlander Shaman, Paladin, and Demon Hunter, as well as Control Priest and Blood Death Knight.

2. **Tempo-Dependent Decks (Category 2)**: These archetypes require a specific mana curve to play cards efficiently and can dominate when curving out correctly. Decks like Treant Druid and Big Rogue, which need to establish a board and then buff it, fall into this category.

3. **Combo-Reliant Archetypes (Category 3)**: These decks hinge on key cards to enable combinations but lack exceptionally fast draw engines. Naga Priest and Dragon Druid, which require a mix of Naga and spells, or Dragon cards in hand respectively, are typical of this category.

4. **Midrange Archetypes with Strong Creatures (Category 4)**: This category thrives on deploying minions with both high attack and health, sustaining board presence to win. Earthen Paladin is a prime example, using minions that can endure and dominate board trades. Their area of effect spells also gives them an advantage against swarm-based strategies (Category 7).

5. **Midrange Value Decks (Category 5)**: These decks aim to overwhelm the opponent with the sheer quality of their cards, not necessarily through a quick victory but through sustained pressure and superior trades.

6. **Diverse Midrange Archetypes (Category 6)**: A mix of decks that don't fit neatly into one archetype but generally win by card value. They often don't have as stark counter relationships and tend to have more even matchups.

7. **Aggro and Snowball Archetypes (Category 7)**: These decks look to establish an early board presence and snowball to victory. They perform well against decks that struggle to clear multiple threats, such as general aggro decks (Category 9).

8. **Value-Oriented Decks with Combo Finishers (Category 8)**: This group features decks that maintain board control with ample resources and are capable of executing a one-turn-kill (OTK) combo in the late stages of the game. Notably, Control Warrior can build up a significant armor stack to unleash a massive hit, while Rainbow Mage is known for its combo potential to achieve an OTK.

9. **Aggressive Paladin Archetypes (Category 9)**: These decks spread the board with multiple threats early on and aim to end the game before the opponent stabilizes.

The analysis reveals that control archetypes (Category 1) are effective against high-value creature decks (Category 4) due to their abundance of removal options. However, they struggle against decks with late-game OTK capabilities (Category 8) because such decks can bypass control strategies with a sudden win condition. Tempo (Category 2) decks falter against stable midrange (Category 4) due to board clears disrupting their momentum, while also struggling against the faster-paced aggro decks (Category 9) that can establish a quicker board presence.

Combination-reliant decks (Category 3) tend to underperform against aggressive Paladin strategies (Category 9) due to the Paladins' ability to conclude games before combos can be assembled. Meanwhile, the snowball potential of decks in Categories 4 and 7 makes them strong against tempo decks but vulnerable to control archetypes with multiple board clears.

Value-oriented decks with combo finishers (Category 8) excel against control decks by circumventing their gradual value game with a sudden win condition, yet they might struggle against decks with large minions that their additional resources can't efficiently counter.

Through the counter table, Hearthstone players can better strategize their deck choices and gameplay, considering the prevalent matchups in the current meta. The table thus serves as a critical tool for players aiming to optimize their strategies and achieve a higher win rate in competitive play.

Table 8: Strength relation accuracies (%) in training and testing for Brawl Stars 2 Heroes.

|  | Training Accuracy | Testing Accuracy |
|---|---|---|
| WinValue | 56.7 | 56.7 |
| PairWin | 80.0 | 74.9 |
| BT | 50.9 | 50.7 |
| NRT | 58.0 | 57.1 |
| NCT M=3 | 58.1 | 57.1 |
| NCT M=9 | 61.5 | 60.2 |
| NCT M=27 | 63.5 | 61.9 |
| NCT M=81 | 65.6 | 63.7 |

| Category1 | Category2 | Category3 | Category4 | Category5 | Category6 | Category7 | Category8 | Category9 |
|---|---|---|---|---|---|---|---|---|
| Highlander Shaman 1.02839 | Treant Druid 1.48915 | Naga Priest 1.20033 | Earthen Paladin 1.36393 | Highlander Hunter 1.27484 | Mining Rogue 1.00962 | Enrage Warrior 0.99643 | Control Mage 1.13171 | Aggro Paladin 1.17278 |
| Highlander Paladin 0.91191 | Big Rogue 1.21318 | Dragon Druid 1.15328 | Pure Paladin 1.02695 | Arcane Hunter 1.12621 | Sludge Warlock 0.91738 | Undead Priest 0.78334 | Mining Warrior 0.98667 | Showdown Paladin 1.00791 |
| Highlander Priest 0.81985 | Cleave Hunter 1.03308 | Mech Rogue 1.00164 |  | Secret Mage 1.05124 | Silver Hand Paladin 0.85971 | Relic Demon Hunter 0.49378 | Control Warrior 0.87570 |  |
| Control Priest 0.75925 | Totem Shaman 0.98028 | Secret Hunter 0.95550 |  | Highlander Blood Death Knight 1.03096 | Elemental Mage 0.84178 | Unholy Death Knight 0.49092 | Rainbow Mage 0.80490 |  |
| Highlander Demon Hunter 0.65419 | Rock 'n' Roll Warrior 0.89733 | Elemental Shaman 0.79008 |  | Highlander Druid 1.00740 | Aggro Demon Hunter 0.81080 |  | Ogre Priest 0.70467 |  |
| Blood Death Knight 0.47515 | Curse Warlock 0.82253 | Mech Paladin 0.77766 |  | Plague Death Knight 0.96237 | Wishing Rogue 0.77634 |  | Highlander Mage 0.63054 |  |
|  | Highlander Warrior 0.56461 | Taunt Warrior 0.64670 |  | Big Shaman 0.90223 | Thaddius Warlock 0.77235 |  | Moonbeam Druid 0.61308 |  |
|  | Thaddius Druid 0.55146 | Secret Rogue 0.62855 |  | Control Warlock 0.88740 | Ogre Rogue 0.70925 |  | Breakfast Hunter 0.37031 |  |

Figure 11: Rating table for Hearthstone decks, showcasing the top eight decks within each category based on their performance in the current meta. The ratings are indicative of the deck's overall strength and potential to win matches within their respective categories.

### A.3.3 Brawl Stars

In games with complex combinations for building compositions like Brawl Stars, it requires much more game knowledge to explain the possible meaning of counter categories from learning. For example, if we consider the composition number of only three heroes in Brawl Stars, there are $C_3^{64} = 41664$ available compositions. It is not easy to check all these compositions and further check their pairwise relationships. According to Table 1, the counter relationship is not clear in Brawl Stars with three heroes and the corresponding game modes and maps. Using NRT can achieve 95.9% average accuracy and adding an $M = 81$ counter table only improves 1.3% extra accuracy. This implies that the scalar rating value already reflects most strength relationships and there is not much significant information that an extra counter table can help with. Thus, we trained another composition setting in Brawl Stars. We only considered two-hero combinations as the compositions, and for each game match, we split it into $C_2^3 \times C_2^3 = 9$ game matches as the training and testing dataset. The corresponding accuracies of this setting are listed in Table 8.

In this setting, the accuracy improvement with the $M = 9$ counter table is 3.5%, implying that some meaningful categories are identified for describing counter relationships. We selected the best model of the

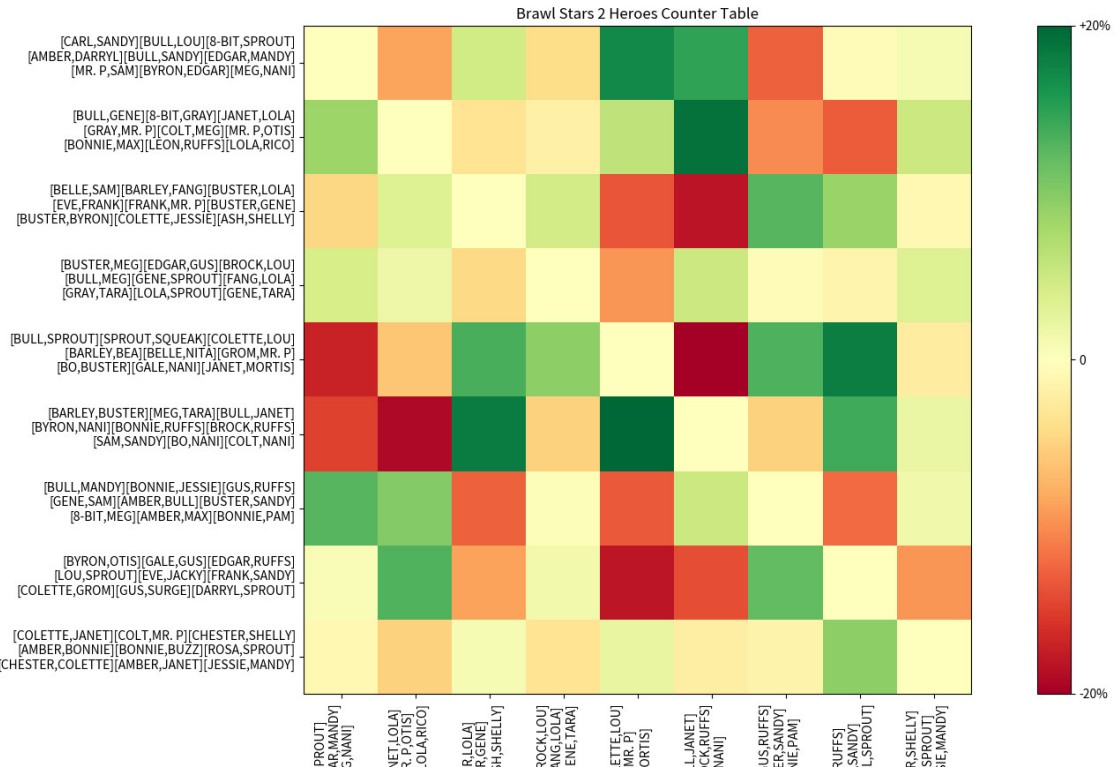

Figure 12: The 9x9 counter table for Brawl Stars under two heroes combinations.

$M = 9$ counter table among five models, where the training accuracy is 62.0% and testing accuracy is 60.2%. The corresponding accuracy improvement from the used NRT model is 4.0% and 3.2%.

Figure 12 shows the result of this counter table. We only list the first nine compositions for each category and try to analyze these compositions with our game knowledge to check whether these compositions also have some clear meaning.

The groups are delineated as follows:

1. **High Mobility and Sustained Output**: Heroes like Sandy, Darryl, and Edgar possess high mobility, while others like 8-Bit, Sprout, and Mr. P provide sustained output.

2. **Versatile Skills and Comprehensive Abilities**: Heroes such as Janet, Lola, and Gray have transformative abilities, allowing them to switch states.

3. **Strong Crowd Control and Suppression**: Heroes like Barley, Frank, and Buster have powerful control and suppression abilities.

4. **High Burst Damage and Support**: Heroes like Edgar, Brock, and Fang offer high burst damage, while others like Gus, Gene, and Tara provide support and assistance.

5. **Long-Range Control and Area Damage**: Heroes like Sprout, Squeak, Barley, and Grom possess long-range control and area damage abilities.

6. **Multi-Skill Coordination**: Heroes like Bonnie, Ruffs, and Byron can coordinate multiple skills, providing sustained output and support.

7. **Sustained Output and Support**: Heroes like Mandy, 8-Bit, and Pam provide sustained output and support.

8. **Versatility and Flexibility**: Heroes like Gale, Gus, and Edgar offer versatility and flexibility, adapting to various tactics.

9. **Quick Burst and Control**: Heroes like Chester, Shelly, and Buzz possess quick burst and control abilities.

Since there are $C_2^{64}$ available compositions and our limited game knowledge, we do not guarantee that these analyses are entirely correct, but we can find some interesting relationships. **High Mobility and Sustained Output** has an advantage over the Long-Range Control and Area Damage category due to their high mobility but may be countered by Sustained Output and Support in modes requiring prolonged engagement. The three categories with versatility: 2, 6, and 8 tend to form a cycle where $2 > 6 > 8 > 2$.

These kinds of ad-hoc explanations may help game designers to change the game mechanisms from a more general aspect. If there is a game that is very hard to explain the counter table or tables diverge when trained from different random seeds, we may need more detailed attributes of the combination elements to summarize a reasonable update direction.

## A.4  Neural Network Architectures and Details

In our study, we employ distinct neural network architectures tailored to the specific requirements of each player-versus-player (PvP) game in our dataset. Below we provide a detailed description of the network designs and input features for each team composition:

- Simple Combination Game: 20-dimensional binary vector representing elements.

- Rock-Paper-Scissors: 3-dimensional one-hot vector for category representation.

- Advanced Combination Game: 23-dimensional vector combining 20-dimensional binary element encoding with 3-dimensional one-hot category encoding.

- Age of Empires II: 45-dimensional one-hot vector for civilization representation.

- Hearthstone: 91-dimensional one-hot vector for deck naming.

- Brawl Stars: 115-dimensional vector encoding the complex dynamics of Trio Modes. This includes a binary encoding for the presence of 64 unique heroes, one-hot encodings for 43 distinct maps plus an indicator for any map not listed, and a similar encoding scheme for the 6 game modes and any unrecorded mode. This encoding captures the essence of team compositions, map strategies, and game modes, essential for predicting match outcomes in Brawl Stars.

- League of Legends: 136-dimensional binary vector for champion representation.

Each network is constructed to handle the dimensionality and characteristics of the input features:

- WinValue Network: Processes individual compositions to output direct win rates.

- PairWin Network: Predicts the pairwise win rate between two compositions.

- BT Network: Implements a linear approximation of the Bradley-Terry model.

- NRT Network: Offers a non-linear approximation of the Bradley-Terry model, capturing complex relationships.

- NCT Network: The neural counter table to consider counter relationships.

For each neural network configuration, we present architectural diagrams that illustrate the structure and flow of data through the network and these network configuration are shared with all games. These visuals serve to complement the textual description and provide an at-a-glance understanding of each model's design.

### A.4.1 Formulations of Strength Relation Methods

For a more formal definition of the methods used in the strength relation classification task, we use the following definitions. For each match outcome, we have two compositions, $A$ and $B$, and a strength relation label with three valid states: weaker/same/stronger. The ground truth of each matchup $A, B$ is determined by the tabular PairWin method, which simply counts the average win value of this composition matchup. For example, if a matchup $A, B$ has 100 game results with 60 wins, 30 losses, and 10 ties, the win value is $x = (60 \times 1.0 + 30 \times 0.0 + 10 \times 0.5)/100 = 0.65$. For $x > 0.501$, the strength relation label is stronger; for $x < 0.499$, the strength relation label is weaker; and for $0.499 \leq x \leq 0.501$, the strength relation label is same.

- WinValue: Given a win value estimator $WinValue_\theta(C)$ of composition $C$ without considering its opponent, if $WinValue_\theta(A) - WinValue_\theta(B) > 0.001$, the prediction is stronger. If $WinValue_\theta(B) - WinValue_\theta(A) > 0.001$, the prediction is weaker. If $|WinValue_\theta(A) - WinValue_\theta(B)| \leq 0.001$, the prediction is same.

- PairWin: Given a win value estimator $PairWin_\theta(C_1, C_2)$ of compositions $C_1$ and $C_2$ in a matchup, if $PairWin_\theta(A, B) > 0.501$, the prediction is stronger. If $PairWin_\theta(A, B) < 0.499$, the prediction is weaker. If $0.499 \leq PairWin_\theta(A, B) \leq 0.501$, the prediction is same.

- BT Network: Given a strength estimator $R_\theta(C)$ for composition $C$, if $\frac{R_\theta(A)}{R_\theta(A) + R_\theta(B)} > 0.501$, the prediction is stronger. If $\frac{R_\theta(A)}{R_\theta(A) + R_\theta(B)} < 0.499$, the prediction is weaker. If $0.499 \leq \frac{R_\theta(A)}{R_\theta(A) + R_\theta(B)} \leq 0.501$, the prediction is same.

- NRT: This method is the same as BT but uses a non-linear activation function in the neural networks.

- NCT Network: Given a strength estimator $R_\theta(C)$ for composition $C$ and an extra counter table $W_\theta(C_1, C_2)$ for adjusting the win value between compositions $C_1$ and $C_2$ in a matchup, if $\frac{R_\theta(A)}{R_\theta(A) + R_\theta(B)} + W_\theta(A, B) > 0.501$, the prediction is stronger. If $\frac{R_\theta(A)}{R_\theta(A) + R_\theta(B)} + W_\theta(A, B) < 0.499$, the prediction is weaker. If $0.499 \leq \frac{R_\theta(A)}{R_\theta(A) + R_\theta(B)} + W_\theta(A, B) \leq 0.501$, the prediction is same.

We count the accuracy of the strength relation label predictions.

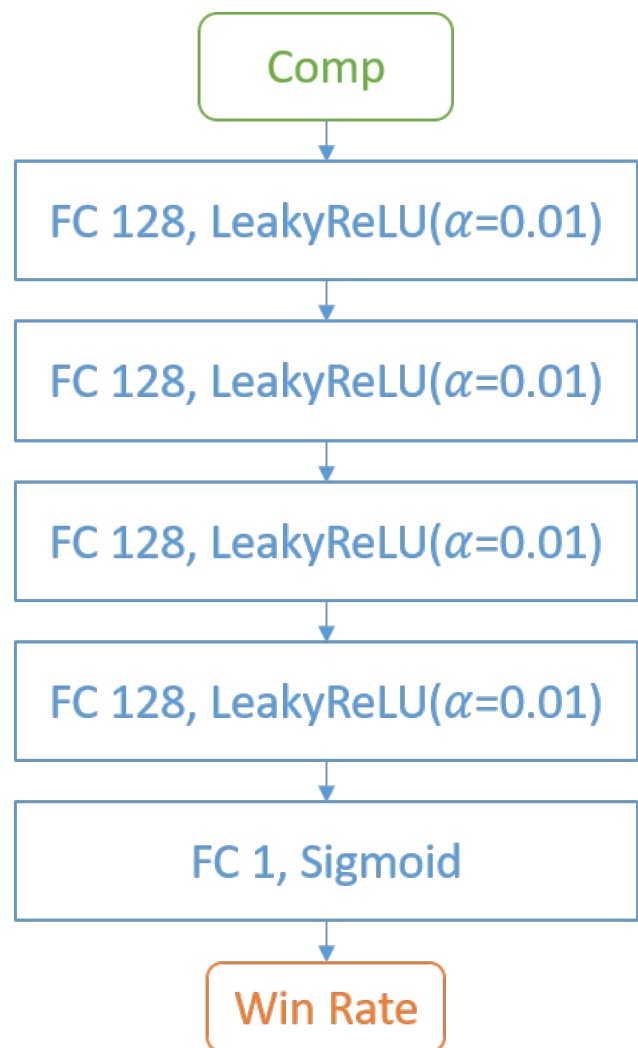

Figure 13: The architecture of the WinValue neural network.

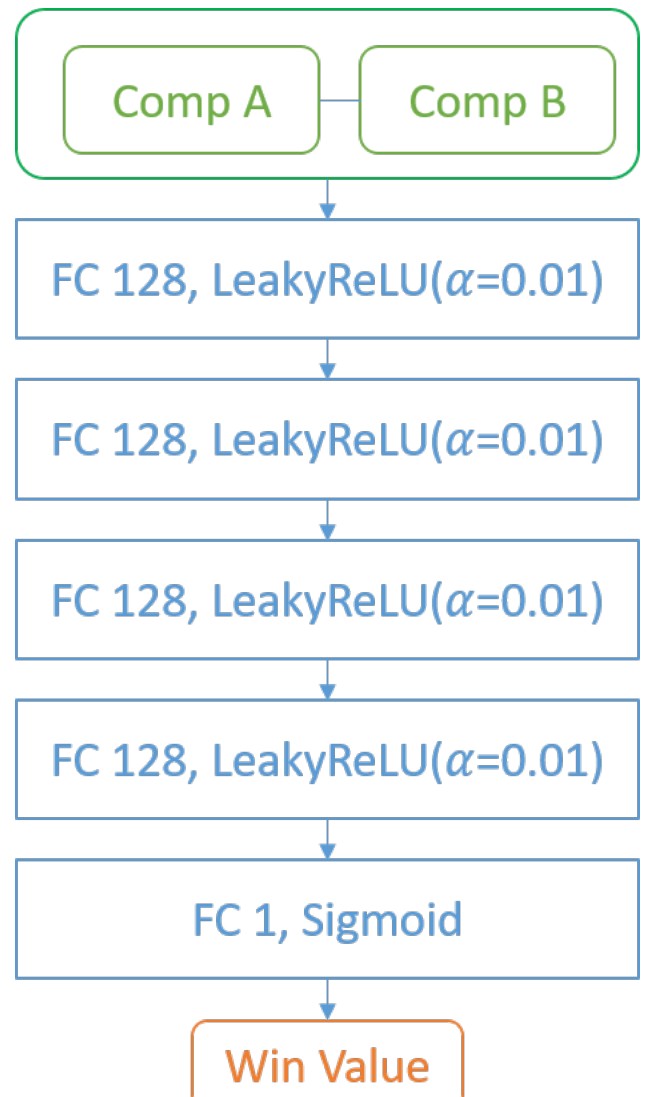

Figure 14: The architecture of the PairWin neural network.

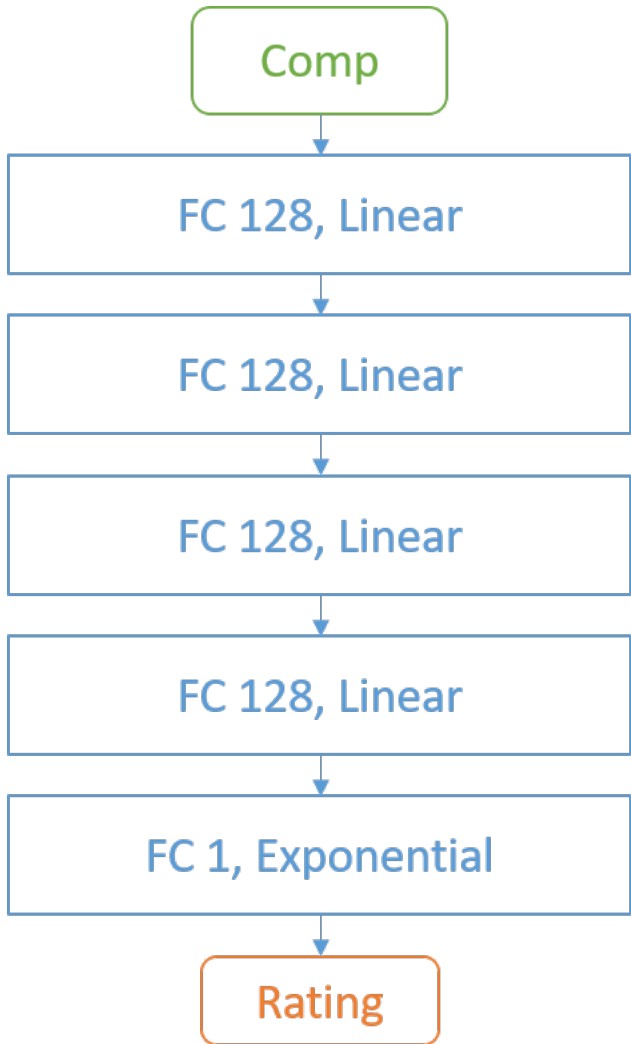

Figure 15: The architecture of the linear Bradley-Terry (BT) model network.

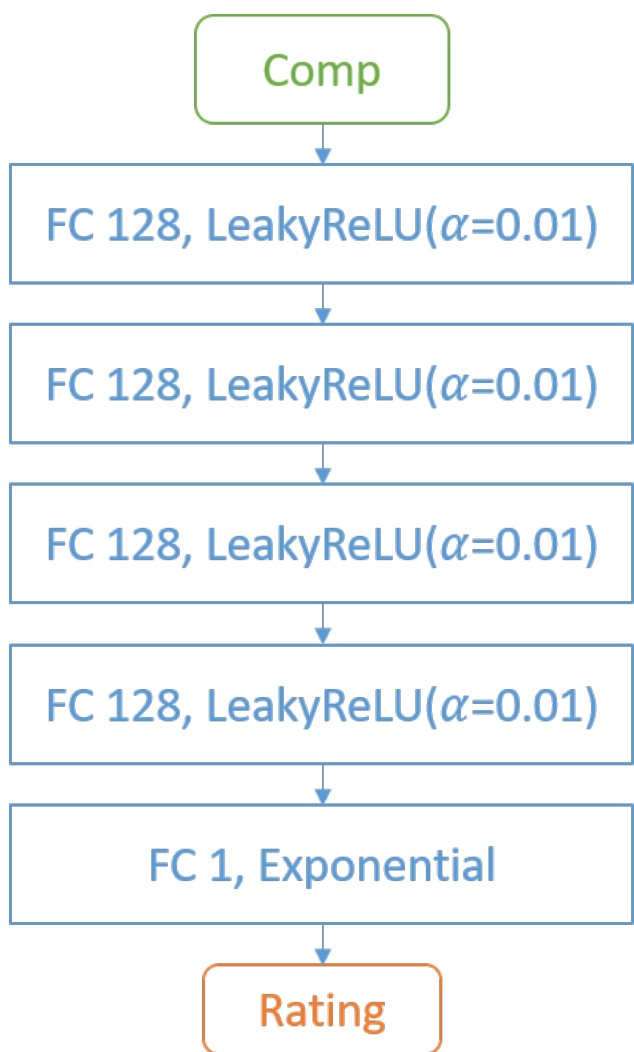

Figure 16: The architecture of the Non-linear Rating Table (NRT) network.

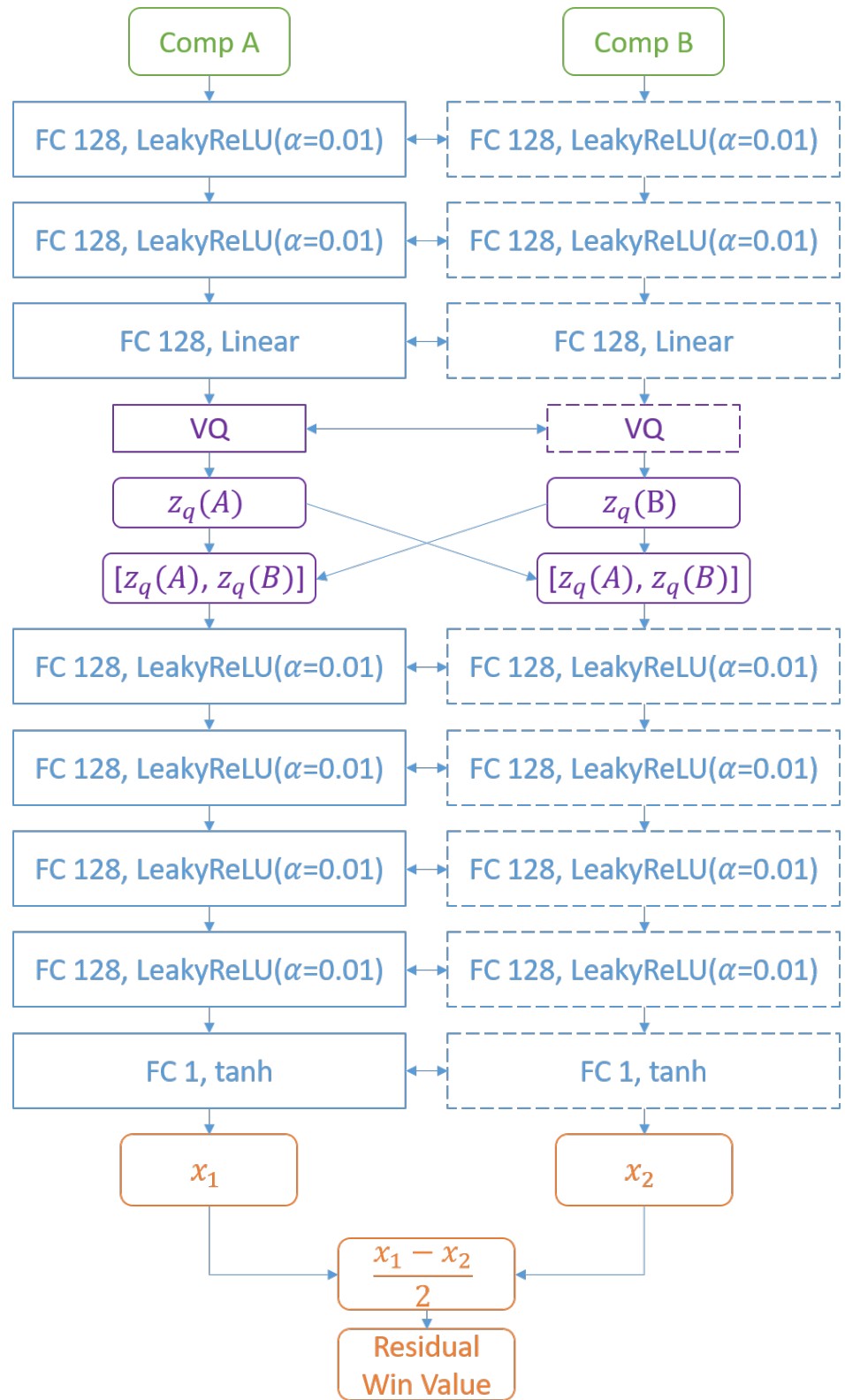

Figure 17: The architecture of the Neural Counter Table (NCT) network.

