# OpenReview forum: "Identifying and Clustering Counter Relationships of Team Compositions in PvP Games for Efficient Balance Analysis"
_TMLR — Accepted by TMLR_

### Review · Reviewer_F1ex · 2024-06-17

**Summary Of Contributions:**

The paper focuses on the problem of `game balance’ that is prevalent in the current video game industry. Although the problem is prevalent; quantifiable metrics are not standardized to measure game balance. Although some measures based on win rates, and entropy of strategies at Nash equilibrium have been proposed in the current literature, the authors claim such methods can be resource-intensive. They propose learning two models namely, the machine learning based Bradley-Terry model and the machine learning based counter table. The authors use these two models to propose two novel metrics namely Top-D Diversity and Top-B Balance which have a better time/space complexity when compared to win rate based metrics. The authors conclude their paper with an in-depth discussion (ablation study) of the proposed methods, suggesting guidelines for tuning the hyperparameters for specific use cases.

**Audience:**

Yes

**Claims And Evidence:**

No

**Requested Changes:**

Requested Changes
1. I would like to see a comparison of compositions considered balanced with (1) win-rate based method (2) entropy of strategies at Nash equilibrium-based method (Pendurkar et a. 2023). (3) proposed methods. That is, highlighting the differences not only between the proposed methods but also with the current literature to support the claim that calculating the proposed metrics is easier (at least in some cases) and better/comparable. These experiments + addressing related work concerns would address my concerns regarding `Claims And Evidence'

Writing Changes
1. 'Treant Druid needs a specific nerf’ The ML audience is not familiar with terms like 'nerf’ and 'buff’. Please define them.
2. In Section 4.1 please mention the composition size per game. (similar to 'k’ and '|M|’ in Pendurkar et al. 2023).
3. Please cite original work on the siamese neural network (page 2 line 2) (Bromley et al. 1993)
4. The section on the counter table model is not very clear, especially, because the background is not appropriately introduced. I would suggest authors add a subsection at the beginning that provides some idea on vector quantization training and defines terms like `codebook’. VQ Mean Loss seems like a contribution for the paper, the differences from the literature should be clearly highlighted.

Questions (I hope authors use my questions to update their paper as well for clarity).
1. What are gameplay policies used (after composition) for each game to collect the data?
2. When should one use Top-D Diversity and when should one use Top-B Balance? Such a discussion is missing. Based on my understanding, it seems Top-B is more useful when using counters (see rock-paper-scissors-fire-water variant used in Pendurkar et al. 2023 which might serve as an easy example to demonstrate this).
3. Are the proposed metrics limited to two-player zero-sum games? What are the challenges if one were to scale them to multi-player games (especially with counter table)? If yes, then please clearly state this while discussing the limitations.
4. Page 2 para 2. ''which counts.. given a tolerant win value gap’’ what do you mean by `tolerant value group’?
5. How is Proposition 5.9 True? Do you assume the ML model for BT rating is accurate? Further, maybe countering c1 might be easier than countering c2 (this might happen if the categories are not appropriately identified). In such cases, 5.9 might not hold. Please clarify your assumptions here.

Bromley, Jane, et al. "Signature verification using a" siamese" time delay neural network." Advances in neural information processing systems 6 (1993).

Minor
1. Just above Definition 2.1 define the target, ‘’domination,’’ -> ‘‘domination’’,
2. Ralf Herbrich, Tom Minka, and Thore Graepel. Trueskilltm: A bayesian skill rating system.
In Advances in Neural Information Processing Systems (NIPS), 2006. (Apologies for bad formatting here) ™ should be in upper case (so should be B in Bayesian).

**Strengths And Weaknesses:**

Strengths
1. The paper presents a good ablation study providing insightful conclusions. Such a study makes it easier for the reader to understand the effect of hyperparameters on performance based on use cases.
2. The paper seems very reproducible as the paper + appendix provides most of the details required to reproduce the results. (In any case, I encourage authors to release their code).
3. The paper discusses literature from both video games as well as ML community, which helps to motivate the importance of the problem being tackled.

Weakness
1. The writing is not clear in some parts, especially the counter table implementation section. See Requested Changes for more details.

2. The paper does not discuss the related work in an appropriate context.
     1. E.g., How do the papers Bowling et al. 2015, and Perolat et al. 2022, support your claim that computing Nash equilibrium is expensive in general? Yes, it might be expensive for the games considered, but not for all games (like rock paper scissors). Pendurkar et al. 2023 present such a study.
     2. Some related (and recent) papers are not discussed (e.g, Hernandez et al 2020., Reis et al. 2023)
     3. The paper cites Pendurkar et al. 2023 in the section that discusses win rate based metrics (Section 1 Introduction). However, the original paper proposed an entropy-based objective orthogonal to win-rate-based metrics (Not cited when you discuss this in the paper). The authors should avoid citing work in the wrong places.
3. Comparison with previous work is missing, making the claims of faster evaluation of metrics weaker. The paper needs an empirical study, of when the proposed metrics are faster/slower to evaluate, and how much these metrics differ from one another (discussing the 'quality' of balance).

Hernandez, Daniel, et al. "Metagame autobalancing for competitive multiplayer games." 2020 IEEE Conference on Games (CoG). IEEE, 2020.

Reis, Simão, et al. "An Adversarial Approach for Automated Pokémon Team Building and Meta-Game Balance." IEEE Transactions on Games (2023).

---

> ### Author Response · Authors · 2024-07-18
>
> Thank you for your detailed comments.
>
> We have uploaded a revised version with changes highlighted in cyan. Please verify if the requested changes have been addressed.
>
> In addition to some minor changes, we have added two new sections to respond to your requests:
>
> 1. We have introduced a new Section 6.3 to discuss the use cases and information that different balance measures can provide. Since the actual effectiveness of a balance measure is very hard to objectively quantify, such as the popularity or revenue of games after using a balance measure, we discuss the intention and information that can be provided for game designers. The choice of which measure to use is ultimately the responsibility of the game designers.
>
> 2. We have added Section 3.2.1 to provide a more detailed introduction to VQ-VAE.
>
> Regarding your questions:
> 1. The gameplay policies are based on human players in online games, and there are no specific players in simple games since those simple games are completed after composition selection.
> 2. Please refer to our new Section 6.3 for detailed discussions.
> 3. The Bradley–Terry model is typically used for two-player zero-sum games. Thus, if a multiplayer game cannot use the Bradley–Terry model to build a rating system, it can be a limitation. However, in our overall approach, we propose first using a rating system to analyze balance based on win rates and then further analyzing balance based on domination relationships. These are not limited to two-player zero-sum games but are strength relationships defined on win rates that can be applied to scenarios with win rates. For the counter table that enables efficient balance analysis, it is built on the residual part that a scalar rating system cannot handle. Since we have only tested on two-team zero-sum games, we do not claim it can be universally applied, but it is still a general idea to identify counter relationships. We have added an additional description in the Broader Impact Statement.
> 4. The tolerant win value gap is discussed in Assumption 5.4.
> 5. Proposition 5.9 is based on Assumption 5.7. If the accuracy of the BT rating or counter category is not acceptable, the assumption does not hold, and Proposition 5.9 is not true. Proposition 5.9 is derived from Assumption 5.7.

---

> > ### Comment · Reviewer_F1ex · 2024-07-25
> > **Official Comment by Reviewer F1ex**
> >
> > Thanks for your detailed response and updating your paper.
> >
> > My concerns regarding questions:
> > 4. Although the definition of tolerant win value gap is used in Section 5, it should still be introduced (at least in lay mans terms) in the introduction if you are using it.
> >
> > Regarding my main concerns:
> > 1. Section 3.2.1 clarifies the basic method. I just have a minor question "How is the codebook (e_x) vectors obtained generally?". This also helps to understand the contribution of the work with respect to baseline work. However, section 3.2.2 is still not clear to me. Especially the intuitations (reasons) why each step is done is not very clear (Equation 5, 8). Can you please elaborate on it?
> >
> > 2. I do not agree with authors omitting comparison to baseline. I agree with authors' comment " choice of which measure to use is ultimately the responsibility of the game designers". However, for designers to make this choice, a study of how they differ is essential. I would like to see in games where having 50% win rates is desirable, the method proposed provides reasonable ratings/counter relationships (this is missing). Further, the discussion in section 6.3 is incorrect. See paragraph 3 in 6.3, the authors claim that "However, this idea has the same limitation as achieving 50% win rates: making all strategies nearly the same" (the idea is Pendurkar et al. 2023 - entropy based balance, etc..). Optimizing the metric proposed makes all strategies ``winnable'' in a sense (and NOT the same). Note, the win rates are not required to be 50% (unlike suggested by authors). This idea allows the counter relationships as suggested by authors. I would like to see comparison of this method with Top-B. I also understand, learning policies might be hard for games like League of Legends, but Kachalsky et al. could be used for hearthstone for example, or optimal strategies could be learnt for the other (simple) games studied. (rock-paper-scissor-fire-water) could also be a nice addition. I also agree with reviewer Reviewer 99kt's comment that the mathematical definitions of the introduced baseline (during rebuttal period) should be added to the paper, as it is confusing.
> >
> > Kachalsky, Ilya, Ilya Zakirzyanov, and Vladimir Ulyantsev. "Applying reinforcement learning and supervised learning techniques to play hearthstone." 2017 16th IEEE International Conference on Machine Learning and Applications (ICMLA). IEEE, 2017.

---

> > > ### Author Response · Authors · 2024-07-26
> > >
> > > Thank you for your reply.
> > >
> > > The codebook vectors in a VQ-VAE are randomly initialized with normal distributions and are trained using the VQ loss, as well as the VQ Mean Loss if our new loss is included. We plan to add the loss terms of VQ-VAE in Section 3.2.1 to further explain the details for those not familiar with VQ-VAE.
> > >
> > > Before our next revision, we would like to ensure that our understanding and your understanding of the entropy-based balance measure (Pendurkar et al. 2023) is aligned.
> > >
> > > Pendurkar et al. proposed the idea of using the entropy of policies approaching Nash equilibrium. There is no measure to validate this method as actually useful for game balance, but it discusses the properties of entropy. The measures used for their bi-level balancing mechanism are based on checking whether the entropy of policies can reach maximum entropy. From the definition and simple mathematical properties of entropy on discrete probability distributions, the distribution of maximum entropy is uniform random, as discussed in Section 3.3 of their paper. Therefore, they added an extra regularization term to prevent this solution, but the target of balance is still defined on maximizing entropy. I am unclear on why you consider our Section 6.3 incorrect, as Pendurkar et al. stated "assigning all items the exact same attribute values leads to a max-entropy MSNE," implying that all strategies have the same strength, which translates to a 50% win rate in symmetric two-team zero-sum games.
> > >
> > > Could you clarify what balance measures you would like to see in comparisons between our method and the entropy-based method? We always need a comparable measure to compare different methods. The entropy-based method uses entropy as the measure and does not have more quantifiable measures since it also discusses methods for changing game parameters, not solely focusing on the balance measure.

---

> > > > ### Comment · Reviewer_F1ex · 2024-07-26
> > > > **Official Comment by Reviewer F1ex**
> > > >
> > > > Thank you for your prompt response.
> > > >
> > > > 1. Regarding VQ-VAE, please update the text as soon as possible. I want to review it and discuss it with other authors before I submit my decision recommendation.
> > > >
> > > > 2. (a) As I stated, I agree there is no well-accepted metric to measure "game balance" that be used to validate, however, several papers have proposed reasonable metrics (as discussed in your submission). The objective proposed by Pendurkar et al. 2023 is a non-convex objective with multiple global optimal solutions. That is, a maximum possible entropy, can be attained with potentially several game meta assignments. This includes one where win rates are the same for each composition (as you suggest). Based on my understanding, Pendurkar et el. 2023, propose adding the initial meta to 1) avoid converging to this solution (I am not sure how many of their solutions converged to this value) 2) adding designer preferences. This helps to converge to an objective that is optimal with respect to entropy + be close to initial meta (see their results, they reach frac_of_max_entropy = 1). Your claim "However, this idea has the same limitation as achieving 50% win rates: making all strategies nearly the same" says that that is the ONLY solution. For example, consider rock-paper-scissor example considered in their paper, according to their objective the original version (without changing pay-off matrix) has maximum possible entropy without having pay-offs 0.5 for all cases (which you claim).  (b) Entropy over strategy of Nash equilibrium is clearly a (quantifiable) metric of balance that Pendurkar et al. 2023. They calculate the metric and try to maximize it with a black-box optimizer. As I previously stated, I would like to see how Top-D and Top-B compare in cases where (1) counter relationships are not present (to understand what are Top-D and Top-B values compared to win-rate-based metric). (2) Where counter relationships are present and comparison with the objective proposed by Pendurkar et al. 2023 (learn a value function like multi arm bandits, and use this entropy over this policy) on some simple games. I understand Pendurkar et al. 2023 might not scale if games are too complex like league of legends. A comparison of one/two domains would suffice.
> > > >
> > > > For 2 (b) I request these experiments to address the question I have "If my game I am trying to balance isn't complex enough; then why should I pick Top-D or Top-B? Are they good enough to measure balance in my game?" that a game designer might have.

---

> > > > > ### Author Response · Authors · 2024-07-26
> > > > >
> > > > > Thank you for your reply.
> > > > >
> > > > > I believe there is still some gap between our understanding and your understanding of the paper by Pendurkar et al. 2023. The policy in Rock-Paper-Scissors for calculating entropy is based on the probability of the used strategy, not the pay-off. In this game, there is only one strategy to reach Nash equilibrium: {1/3, 1/3, 1/3}. In the paper by Pendurkar et al. 2023, a policy approaching Nash equilibrium may not be uniformly random, but the direction for improving entropy is towards uniform randomness.
> > > > >
> > > > > In discrete probability, there is only one distribution that can reach maximum entropy, and that is uniform probability. Please refer to information theory or the Wikipedia page on entropy (https://en.wikipedia.org/wiki/Entropy_(information_theory)).
> > > > >
> > > > > Only if my understanding of the calculation of entropy is incorrect (not the strategy) or the math of information theory is wrong would it make the maximum entropy not uniform random. I am not sure which part I misunderstand, or perhaps other reviewers or the action editor can help find a policy in Rock-Paper-Scissors that reaches Nash equilibrium but is not uniformly random.

---

> > > > > ### Author Response · Authors · 2024-07-27
> > > > >
> > > > > To further clarify our understanding of the method, do you consider the win rate to be the win rate in the pairwise case (pay-off)? In our Section 6.3, the concept of same strength is based on the scalar rating value, similar to the Bradley-Terry Model, which does not consider specific opponents. Therefore, it is not the pay-off. If all strategies have the same strength, it implies that all compositions have the same strength rating value, and their average win rate, when considering only their scalar rating, would be 50% since they have the same strength.
> > > > >
> > > > > Additionally, regarding the requested experiments, do you mean comparing the application of balance measures for performing balance updates (which is not the original scope of our paper) and evaluating what results the measures approve for passing the balance test? If this is the case, we may examine different pay-off scenarios in Rock-Paper-Scissors to check its balance measure. There will be a clear case where the entropy-based measure agrees that all matchups can be ties due to a uniform strategy on all tie matchups still being Nash equilibrium and reaching maximum entropy. Meanwhile, using Top-B Balance would report 1 due to vector quantization being deterministic and all three compositions sharing the same counter category (0 win value adjustment), thus there is only one category, unlike the original version of Rock-Paper-Scissors which has three.

---

### Review · Reviewer_6sdD · 2024-06-30

**Summary Of Contributions:**

The paper introduces methodologies for comparing the strengths of team compositions, and measuring the degree of balance between compositions in games, with an emphasis on video games. The introduced methods aim to address weaknesses of approaches based on analysis of win-rates by taking into account counter relationships, and using neural representations in embedding space rather to avoid high cost enumeration over the combinatorial space of compositions.

The paper makes contributions in several areas:
* The design and training of neural networks to represent team composition strength and counter relationships. This entails encoding team compositions, and using vector quantization to cluster them into a set number of categories, to reduce the complexity of analysing pairs. The authors address the known problem of low codebook utilisation in vector quantization by introducing a new loss term to encourage all codes to move towards the embedded composition, rather than only the nearest neighbour.
* Introduction of metrics to quantify the level of balance in a game, in terms of the number of viable compositions. These metrics build on the learned counter relationships to reduce the algorithmic complexity to scale with the number of categories, rather than the number of possible compositions.
* The paper includes analysis of case studies using the learned ratings, and counter relationships in order to validate the learned relationships with human subject matter expert knowledge, and provide examples of how the knowledge can be used to inform interventions to improve the gaming experience.

**Audience:**

Yes

**Broader Impact Concerns:**

The included broader impact statement adequately covers concerns.

**Claims And Evidence:**

No

**Requested Changes:**

We request additional information and clarification in the following areas:

* Strengthen
    * In section 4.1, counts of the number of instances, and possible compositions are listed for each of the datasets. It would also be informative to include the number of unique compositions represented in the datasets. This can highlight the differences between games like AoE II where a composition is defined by one of 45 factions, all of which are represented, and League of Legends where the composition space is combinatorial and much larger than the number of instances.
    * In eqn. 3, the training objective for the NRT method is described as the mean squared error between the match outcome and the predicted match outcome probability. It is more common in Bradley-Terry inspired models to minimise cross entropy due to the probabilistic nature of the variables. An explanation for the choice of mean squared error would strengthen the understanding.
    * In order to demonstrate the effectiveness of the clustering on the high dimensional compositions spaces, authors may consider adding a case study on a game with combinatorial composition space such as League of Legends or Brawl Stars. While full analysis of all comps is not feasible, distributional level statistics on the cluster membership, and champion cluster membership could be enlightening.
    * Authors should provide definitions of gaming terminologies such as “buffing” and “nerfing.”

* Critical
    * An explanation for the train/test discrepancy and low test accuracy for the League of Legends experiments is necessary.

**Strengths And Weaknesses:**

# Strengths
* The introduction and application of the dominance based metrics to measure strength and balance in combinatorial spaces with transitivity demonstrates value towards understanding and improving balance in games.
* The NCT method of learning counter relationships combines several good ideas.
    * The learning signal is derived from the “residual win value”, identifying that the difference between actual outcome and predicted outcome based on scalar strength contains the composition counter information.
    * The methods use neural encoding in general and  vector quantization in particular to represent the large composition space in a much smaller and more manageable space with interpretable membership qualities.




# Weaknesses
* In Tables 1 and 2, several games exhibit large discrepancies between the train and test accuracies which usually indicates overfitting. This is not addressed in the paper. In particular, the proposed NCT method achieves >90% train accuracy on League of legends with only a 51% test accuracy indicating almost no gain over random guessing.
* The descriptions of the baseline models of WinValue, PairWin, and Bradley-Terry are not described clearly in the same way as NRT and NCT. Additional information is given in the Appendix but this is not mentioned in the baseline descriptions. WinValue and BT appear nearly identical to NRT with WinValue having a difference in the label (win rate vs rating), and BT lacking non-linear activation functions in the network layers. These may be more clearly represented as an ablation of variants of the NRT method.
* The paper introduces a new method to address low codebook utilisation in vector quantization. While they mention other methods which address this problem, they do not provide a comparison of VQ Mean Loss to any other method.
* In the case studies, analysis of game balance is made under different experimental settings such as the number of categories. There are some instances of unclear and apparent contradiction in conclusions. In the Hearthstone study authors state that “From Top-D diversity… the balance is not good”, and “If we extend the counter table to M = 9, … the balance is surprisingly good”, and “When we use a larger counter table, M = 27 or M = 81, the game is balanced enough.” All of these experiments are done on the same dataset representing the game in the same state of balance. The descriptions are referring to how the metrics portray balance differently under different settings, not real changes to the balance of the game.
* In the AoE II and Hearthstone case studies, experiments are run with M=81 indicating 81 composition categories. However in the datasets used there are only 45 comps in AoE II and 58 for Hearthstone after filtering to comps with more than 100 appearances. This means there are more possible categories than compositions in these experiments which runs contrary to the motivation of reducing the comparison space.

---

> ### Author Response · Authors · 2024-07-18
>
> Thank you for your comments and suggestions.
>
> It is true that the models for League of Legends exhibit overfitting. We have added additional explanations in Section 4.2 to address this issue. Furthermore, we have refined Section 4.1 to clarify the number of compositions more explicitly.
>
> Regarding the choice of loss function, we have provided an explanation in Section 3.1.
>
> We also plan to expand the description in Appendix A.3 to include Brawl Stars. However, before we include this part, we would like to ask if the case study you mentioned should focus on verifying the effectiveness of clustering based on game knowledge or also include the balance measure study discussed in the main paper Section 6.
>
> For other minor revisions, please refer to the cyan-colored text in the revised version.
>
> Thank you again for your detailed feedback.

---

### Review · Reviewer_99kt · 2024-07-10

**Summary Of Contributions:**

Strategies in strategic interactions can be given scalar ratings that capture in some sense their “strength”. However, scalar ratings cannot predict cycles that may appear in a strategic interaction, and instead can only capture some transitive relationship between the strategies. Knowing if a strategy dominates another from ratings is useful, but one cannot be sure that there are no cycles.

This paper focuses on PvP games for the “strategic interactions” and team compositions for the “strategies”. It proposes a multidimensional rating that attempts to provide a more accurate win-rate prediction compared to single dimensional approaches (averages, Elo, etc.). It is based on finding a base win-rate and adding a residual network to further hone the win-rate. Importantly, this residual has smaller dimensionality, because it clusters compositions allowing the approach to scale. To demonstrate the utility of their rating method they propose balance measures which may help game designers better balance their compositions so that each composition fits some sort of niche (and is not strictly dominated by all other compositions).

**Audience:**

Yes

**Broader Impact Concerns:**

I have no ethics concerns.

**Claims And Evidence:**

No

**Requested Changes:**

[Important] Please add simple baselines (average, Elo, mElo) in Table 1.

[Important] The authors need to be careful when talking about space complexity. Significant information will be stored in the weights of the neural network. Having approximately 6x128x128 weights in a neural network to rate compositions with N << 6x128x128 and describing it as O(N) will raise a few eyebrows. Please reword or de-emphasise the space complexity. Or only mention it in the O(M^2) part - where it is better motivated.

[Important] Furthermore the authors need to be careful with time complexity. A lot of network pre-training is involved and the neural network weights cannot be used for different datasets. Therefore it is misleading to say that it only takes O(N+M^3) to do top B balance, because it involves so much pre-training.

There are a bunch of definitions, assumptions, propositions and lemmas scattered in the text. I did not find this format easy to follow. Is Lemma 5.6 a lemma, for example? The proofs could be more precise.

[Important] The use of neural networks is under-motivated. The main advantage of using a function approximator is to exploit either its a) generalization capabilities, b) representation efficiencies, or c) both. I guess the LOL dataset does have many more possible compositions than examples, but the rest do not, and therefore do not utilize generalization. And for many of the datasets, computing statistics over the whole dataset requires fewer parameters than the number of weights in the network. I think it is particularly important to address this question because the authors use the network to enumerate the O(N) statistics after training it. Why not just compute the exact statistics directly. (I get that there is an implicit defense of this for the O(M^2) residual part, but I think this needs to be addressed directly in the text).

The method proposed only works with symmetric, 2-player, zero-sum, games. The team games are analyzed as two-player games, where each team is a player. This is not a weakness, as many other rating schemes have this limitation, but I think it should be mentioned in the text.

The “secret sauce” of the paper seems to be the clustering part which, using a differentiable model, clusters the compositions to allow one to deal with statistics that are only O(M^2). This allows one to scale. I *like* this scalability property - I think that is the interesting part of the paper. I suppose I would have preferred the paper to focus on selling that part rather than distracting the readers with lemmas that do not feel tight, over-emphasising the importance of neural network training, or being a bit misleading with complexity.

I hope this review is helpful and improves your paper.

**Strengths And Weaknesses:**

Strengths

1. Single-dimensional ratings are indeed problematic and identifying cycles is important. This work is tackling an interesting problem.
2. Attempts to tackle the scale problem using category clustering.
3. I enjoyed the case studies of real games in the appendix.

Weaknesses

1. There are no non-neural-network baselines. I would have liked to see baselines against simple average, Elo, and multidimensional Elo.
2. I feel the space/time complexity analysis is misleading or at least should be clarified.
3. I feel the wrong parts of the paper are emphasized (clarified below).

Minor Comments

1. “MOBA” is undefined in the abstract.
2. “Beginning with 3 to capture basic cycle dominance…”. Cycles can appear with just two strategies (matching pennies). I suppose this  paper mainly focuses on symmetric two-player zero-sum which requires three actions.
3. Figure 2: Some boxes are dashed while others are not.
4. Lemma 5.6 - is this a Lemma?
5. I’m unsure of the distinction between “​​PairWin” and “WinValue”.

---

> ### Author Response · Authors · 2024-07-18
>
> Thank you for your detailed comments.
>
> For minor changes, please refer to the cyan-colored text in the revised version.
>
> Lemma 5.6 is derived from Definition 5.3, Assumption 5.1, Assumption 5.4, Assumption 5.5, and Proposition 2.2. If the corresponding conditions are true, Lemma 5.6 is always true.
>
> Regarding the distinction between "PairWin" and "WinValue", we have described it in Section 4.2. To clarify, WinValue is a function with only one composition as the input, while PairWin considers two compositions from each side. Thus, WinValue may not be accurate enough since it does not consider pairwise matchup conditions.
>
> In this paper, we use neural networks to approximate those compositions or matchups that may not appear in the training data. We can add some baselines in a future revision with tabular approximations like simple win value, pairwise win value, and Elo rating. However, unseen compositions or matchups will be assigned an unknown result, which will always lead to incorrect strength relation predictions.
>
> Regarding space complexity, our discussion focuses on performing balance analysis, not including the data collection and preparation of necessary tools. The space complexity is relevant for generating balance reports, for instance, when checking the corresponding ratings and categories of each composition, but we would not examine the weight values of neural network models.
>
> After reviewing your comments, we realized there might be a misunderstanding or misalignment with our intended task (game balance) since the Summary of Contributions in the second paragraph does not accurately reflect our approach. We are not proposing a multidimensional rating system; instead, we use a scalar rating and a category, which is not the same as a multidimensional rating. This paper is focused on game balance, not on developing a common rating system for matchmaking.
>
> We do not plan to change the main topic from game balance to clustering counter categories for other potential applications. If you do not find game balance to be a valuable problem or consider our contributions to be minor, please recommend rejection and provide the corresponding reasons.

---

> ### Comment · Reviewer_99kt · 2024-07-25
>
> > Regarding the distinction between "PairWin" and "WinValue", we have described it in Section 4.2. To clarify, WinValue is a function with only one composition as the input, while PairWin considers two compositions from each side. Thus, WinValue may not be accurate enough since it does not consider pairwise matchup conditions.
>
> The definitions are still a bit circular, and remain unconnected to the losses explained in previous sections. Expressing these quantities mathematically would be clearer. E.g.:
>
> WinValueIndicator =
>
> [  +1  if WinValueNet_theta(C_1) - WinValueNet_theta(C_2) > 0.1
>
> [  -1  if WinValueNet_theta(C_1) - WinValueNet_theta(C_2) < -0.1
>
> [   0  else
>
> Where WinValue_theta is trained on *blah blah* loss.
>
> WinValueMetric := MEAN(WinValueIndicator == TrueWinIndicator)
>
> And PairWinValue_theta(C_1, C_2) in [0, 1] is trained *blah blah* loss.
>
> PairWinValueIndicator =
>
> [  +1  if PairWinValue_theta(C_1, C_2) > 0.1
>
> [  -1  if PairWinValue_theta(C_1, C_2) < -0.1
>
> [   0  else
>
> PairWinValueMetric := MEAN(PairWinValueIndicator == TruePairWinIndicator)
>
> Or something like this.
>
> > In this paper, we use neural networks to approximate those compositions or matchups that may not appear in the training data. We can add some baselines in a future revision with tabular approximations like simple win value, pairwise win value, and Elo rating. However, unseen compositions or matchups will be assigned an unknown result, which will always lead to incorrect strength relation predictions.
>
> Sure, nevertheless I think proving that the networks give sensible (similar than tabular) results on in-distribution data is important. I see that you have added this data, thank you.
>
> > Regarding space complexity, our discussion focuses on performing balance analysis, not including the data collection and preparation of necessary tools. The space complexity is relevant for generating balance reports, for instance, when checking the corresponding ratings and categories of each composition, but we would not examine the weight values of neural network models.
>
> Ok, I take your point. On my first read through I misunderstood the significance of the big-O discussions.
>
> > After reviewing your comments, we realized there might be a misunderstanding or misalignment with our intended task (game balance) since the Summary of Contributions in the second paragraph does not accurately reflect our approach. We are not proposing a multidimensional rating system; instead, we use a scalar rating and a category, which is not the same as a multidimensional rating. This paper is focused on game balance, not on developing a common rating system for matchmaking. We do not plan to change the main topic from game balance to clustering counter categories for other potential applications. If you do not find game balance to be a valuable problem or consider our contributions to be minor, please recommend rejection and provide the corresponding reasons.
>
> I think the application is interesting and I do not wish you to change the topic. My comments were mainly around the framing and positioning of the technique.

---

> > ### Author Response · Authors · 2024-07-26
> >
> > Thanks for your reply.
> >
> > We will add the following explanations in our next revision in Section A.4.1. Does this adequately address the problems in the definition of methods for the strength relation classification task?
> >
> > For a more formal definition of the methods used in the strength relation classification task, we use the following definitions. For each match outcome, we have two compositions, $A$ and $B$, and a strength relation label with three valid states: weaker/same/stronger. The ground truth of each matchup $A,B$ is determined by the tabular PairWin method, which simply counts the average win value of this composition matchup. For example, if a matchup $A,B$ has 100 game results with 60 wins, 30 losses, and 10 ties, the win value is $x = (60 \times 1.0 + 30 \times 0.0 + 10 \times 0.5)/100 = 0.65$. For $x > 0.501$, the strength relation label is stronger; for $x < 0.499$, the strength relation label is weaker; and for $0.499 \leq x \leq 0.501$, the strength relation label is same.
> >
> > * WinValue: Given a win value estimator $WinValue_{\theta}(C)$ of composition $C$ without considering its opponent, if $WinValue_{\theta}(A) - WinValue_{\theta}(B) > 0.001$, the prediction is stronger. If $WinValue_{\theta}(B) - WinValue_{\theta}(A) > 0.001$, the prediction is weaker. If $|WinValue_{\theta}(A) - WinValue_{\theta}(B)| \leq 0.001$, the prediction is same.
> > * PairWin: Given a win value estimator $PairWin_{\theta}(C_1,C_2)$ of compositions $C_1$ and $C_2$ in a matchup, if $PairWin_{\theta}(A,B) > 0.501$, the prediction is stronger. If $PairWin_{\theta}(A,B) < 0.499$, the prediction is weaker. If $0.499 \leq PairWin_{\theta}(A,B) \leq 0.501$, the prediction is same.
> > * BT Network: Given a strength estimator $R_{\theta}(C)$ for composition $C$, if $\frac{R_{\theta}(A)}{R_{\theta}(A) + R_{\theta}(B)} > 0.501$, the prediction is stronger. If $\frac{R_{\theta}(A)}{R_{\theta}(A) + R_{\theta}(B)} < 0.499$, the prediction is weaker. If $0.499 \leq \frac{R_{\theta}(A)}{R_{\theta}(A) + R_{\theta}(B)} \leq 0.501$, the prediction is same.
> > * NRT: This method is the same as BT but uses a non-linear activation function in the neural networks.
> > * NCT Network: Given a strength estimator $R_{\theta}(C)$ for composition $C$ and an extra counter table $W_{\theta}(C_1,C_2)$ for adjusting the win value between compositions $C_1$ and $C_2$ in a matchup, if $\frac{R_{\theta}(A)}{R_{\theta}(A) + R_{\theta}(B)} + W_{\theta}(A,B) > 0.501$, the prediction is stronger. If $\frac{R_{\theta}(A)}{R_{\theta}(A) + R_{\theta}(B)} + W_{\theta}(A,B) < 0.499$, the prediction is weaker. If $0.499 \leq \frac{R_{\theta}(A)}{R_{\theta}(A) + R_{\theta}(B)} + W_{\theta}(A,B) \leq 0.501$, the prediction is same.
> >
> > We count the accuracy of the strength relation label predictions.

---

### Author Response · Authors · 2024-07-18

Thank you to all reviewers for their comments to improve this paper.
We have uploaded a revision, including new sections that address the requested discussions and minor suggestions.
The modified parts are highlighted in cyan.
We also plan to upload another revision to include the counter category for Brawl Stars to check the complex combinations of heroes, as well as some requested additional baselines. If there are any other important parts that need to be addressed, please notify us again.

---

### Author Response · Authors · 2024-07-24
**Revision2**

Dear Reviewers and Action Editor,

We have uploaded a new revision with magenta highlights. This revision includes the following major changes along with some minor polishing to improve presentation:

1. We have clarified the complexity focus in the latter part of the introduction.
2. We have clarified our application to two-team zero-sum games.
3. We have added a new Section 4.4, which includes some tabular baselines without neural networks for the strength relation prediction task. The results demonstrate the necessity of using neural networks in our applications.
4. We have added a new Appendix 3.4 to discuss a more complex composition case in Brawl Stars with two-hero combinations.

Thank you for your insightful comments and suggestions.

---

> ### Comment · Reviewer_99kt · 2024-07-25
>
> Thank you for these additions.

---

### Author Response · Authors · 2024-07-29
**Revision 3**

Dear reviewers and action editor,

We have uploaded a new revision with orange highlights. This revision includes the following updates:

1. Detailed Explanation of Strength Relation Classification Methods: We have expanded Section A.4.1 to include more formal definitions and explanations of the methods used in the strength relation classification task.

2. Clarification of Tolerant Win Value Gap in Introduction: In the introduction, we have clarified the concept of a tolerant win value gap and its importance in defining playable compositions.

3. VQ-VAE Training Process: We have added detailed descriptions of the VQ-VAE training process, including the loss functions and training steps, to Section 3.2.1 for better understanding.

4. Further Discussion on Balance Measures with Examples: We have expanded Section 6.3 to include a more in-depth discussion on different types of balance measures. Additionally, we have provided examples to illustrate the advantages of our proposed measures over traditional ones.

These changes aim to address the reviewers' comments and enhance the clarity and comprehensiveness of our paper. We appreciate your continued feedback and look forward to your thoughts on this revision.

---

### Decision · Action_Editor_tKR7 · 2024-08-11

**Recommendation:** Accept with minor revision

**Comment:**

All three reviewers (as well as myself) agree on the relevance of this work to (a subset of) the TMLR audience.

We also mostly agree on the claims of the paper being supported by the experiments and discussion, just with one potential concern remaining regarding the discussion and/or comparison to other types of measures of game balance proposed in prior research. In light of this, **I have the following suggestion for what I think would be a very minor revision** (where I invite the authors to also explain if they feel that this would be a bad change to make):

At the end of the third paragraph of Section 6.3, there is currently one sentence alluding to the regularization term that was proposed by Pendurkar et al. (2023) for their entropy-based approach, which, without such a term, could suffer from the existence of undesirable solutions that maximise entropy simply by making every behave exactly the same (which would be a "balanced" solution with equal win percentages for all comps, but also uninteresting in terms of player experience). More specifically (not yet discussed in your paper), this regularization term punishes the balancing procedure of Pendurkar et al. (2023) for moving game parameters too far away from the initial parameters, where the initial parameters are assumed to have been set by the game designers. This initial game design is likely not solely informed by wishing to produce a balanced game, but also the wish to ensure that certain comps/decks/characters fit specifically designed playstyles, match a certain theme/personality of a character, and so on. My impression is that such considerations would not solely be relevant to the specific algorithm of Pendurkar et al. (2023), but could be a relevant consideration (maybe: a constraint) for **any** measure of game balance. Regardless of which measure of balance a game designer would use (any of the ones proposed in this paper, or others discussed as related work in Section 6.3 of this paper), there would often be a constraint not to drift too far from the original game design.

I would suggest that it may be worth adding a few (e.g., two or three) sentences to discuss these ideas in the larger discussion of Section 6.3.

---

Aside from the above suggestion, I would like to remark to the authors:
- Please do not forget to turn all text back to the regular black text for the camera-ready version.
- Throughout the Appendix, the letter "x" is used in a few places where $\times$ (typeset using `$\times$`) would look much nicer: specifically, when discussing "9x9 counter tables" for Age of Empires 2 and Hearthstone.

**Audience:**

At the very least, this paper will be of interest to a subset of the TMLR audience, in particular: those interested in studying dynamics such as countering relationships between compositions/decks/etc. in (competitive) video games and analysing, quantifying, or improving their balance.

**Claims And Evidence:**

Claims are backed up by (1) experiments in domains including toy domains (e.g., Rock-Paper-Scissors) as well as large datasets of popular video games (e.g., Hearthstone, Brawl Stars, League of Legends, ...), and (2) discussion.

---

> ### Author Response · Authors · 2024-08-17
>
> Thank you to all the reviewers and the action editor for your acknowledgment and feedback.
>
> We are currently preparing the code release and presentation video for the camera-ready version, which will take a few days.
> This includes an internal presentation to our lab members to gather questions and feedback for the presentation video.
>
> Regarding the concerns and discussion about the trade-off between improving game balance and maintaining the original design parameters, we have uploaded a revision that further explores this topic (Section 6.3, still highlighted in orange, with previous changes reverted to black text).
>
> Here is a brief overview of the new explanation:
> In game design, balance is just one of many factors considered [1]. Designers sometimes compromise on balance in favor of more important themes or features that enhance game experience or playstyle diversity. The regularization term in the entropy-based balance measure is an example of implementing this trade-off, similar to how the tolerance gap functions in our Top-D Diversity measure.
>
> For reference [1], we plan to cite The Art of Game Design: A Book of Lenses. We may also include some concrete factors in the final version if it can help clarify and strengthen the discussion.
>
> Additionally, we have corrected the "x" in 9x9. If there are any other issues we might have missed, please let us know so we can address them.
> Also, if you have any questions or points that you would like to see highlighted in our presentation slides for future readers, please feel free to provide them here.

---

> > ### Comment · Action_Editor_tKR7 · 2024-08-17
> >
> > Thanks for the update. The minor revision looks good to me.